# AgentTTS: Large Language Model Agent for Test-time Compute-optimal Scaling Strategy in Complex Tasks

**Fali Wang**[1]*, **Hui Liu**[2], **Zhenwei Dai**[2], **Jingying Zeng**[2], **Zhiwei Zhang**[1], **Zongyu Wu**[1],
**Chen Luo**[2], **Zhen Li**[2], **Xianfeng Tang**[2], **Qi He**[2], **Suhang Wang**[1]

[1]The Pennsylvania State University, University Park, PA, USA
[2]Amazon, Palo Alto, CA, USA
{fqw5095,zbz5349,zzw5373,szw494}@psu.edu
{liunhu,zejingyi,zwdai,cheluo,xianft}@amazon.com

## Abstract

Test-time scaling (TTS) enhances the performance of large language models (LLMs) by allocating additional compute resources during inference. However, existing research primarily investigates TTS in single-stage tasks; while many real-world problems are multi-stage complex tasks, composed of a sequence of heterogeneous subtasks with each subtask requires LLM of specific capability. Therefore, we study a novel problem: the test-time compute-optimal scaling in multi-stage complex tasks, aiming to select suitable models and allocate budgets per subtask to maximize overall performance. TTS in multi-stage tasks introduces two fundamental challenges: (i) The combinatorial search space of model and budget allocations, combined with the high cost of inference, makes brute-force search impractical. (ii) The optimal model and budget allocations across subtasks are interdependent, increasing the complexity of the compute-optimal search. To address this gap, we conduct extensive pilot experiments on four tasks across six datasets, deriving three empirical insights characterizing the behavior of LLMs in multi-stage complex tasks. Informed by these insights, we propose AgentTTS, an LLM-agent-based framework that autonomously searches for compute-optimal allocations through iterative feedback-driven interactions with the execution environment. Experimental results demonstrate that AgentTTS significantly outperforms traditional and other LLM-based baselines in search efficiency, and shows improved robustness to varying training set sizes and enhanced interpretability. [2]

## 1 Introduction

Test-time scaling (TTS), which refers to allocating additional computational resources during inference, has shown promising results in improving the performance of large language models (LLMs) [2, 45, 55, 33, 19, 54, 26]. For example, Brown et al. [2] scaled inference compute via repeated sampling of candidate solutions, using a verifier to select the best prediction, enabling a weak small model to outperform single-sample state-of-the-art models. Despite its effectiveness, existing methods are primarily designed for single-stage tasks, where the model performs a single function, such as mathematical problem solving [45, 2, 55, 33, 19, 26, 14] or code generation [2, 34]. However, many real-world applications involve *multi-stage complex tasks*, for which compute allocation remains underexplored. These tasks involve a sequential execution of heterogeneous subtasks to accomplish complex objectives. Representative examples include retrieval-then-generation QA systems [16], waterfall-style software development (comprising requirement analysis, system design,

---

*Work done during an internship at Amazon.
[2]Code link: https://github.com/FairyFali/AgentTTS/

39th Conference on Neural Information Processing Systems (NeurIPS 2025).

coding, and testing) [29], and multi-agent task automation (such as task decomposition, tool selection, and parameter prediction) [32]. Each subtask within a multi-stage workflow often requires a model with specific capabilities [13]. For example, in a retrieval-then-generation QA task, retrieval benefits from large models with strong long-context understanding, while generation can achieve competitive performance using smaller models with repeated sampling (see Fig. 1(a)(b)). Therefore, multi-stage tasks demand models with diverse types and levels of capabilities, challenging existing TTS methods.

To bridge this gap, we formulate and study a novel problem: **test-time compute-optimal scaling in multi-stage complex tasks**: for a complex task composed of a sequence of subtasks and each subtask has a set of candidate models to choose from, given a total compute budget, the objective is to select the appropriate model and allocate the compute budget for each subtask to maximize overall task performance. This problem presents two major challenges. **(i)** The search space is large due to the combinatorial choices of models and budget allocations across subtasks. For instance, in software development with three subtasks and two model options each (3B and 70B), the number of configurations can reach up to $10^6$ (see Appendix A.1 for a calculation example), with each configuration requiring time-consuming inference (often several hours), rendering brute-force search impractical. **(ii)** Subtasks are not independent. The compute allocation for one subtask affects the performance and budget requirements of others. As shown in Fig. 1(b-d), high-quality retrieval can significantly reduce the compute required by the downstream generation subtask to achieve peak performance. In contrast, poor retrieval quality necessitates increased computation for generation, either through additional sampling or the use of larger models, to compensate for degraded input quality. These challenges highlight the need for an efficient search strategy that can handle the large and interdependent search space.

To address these challenges, we first conduct preliminary experiments to characterize the behavior of LLMs under test-time scaling in multi-stage tasks. From experiments, we derive three generalizable insights across four task types: (1) Different subtasks exhibit distinct preferences between large and small models; (2) Increasing test-time compute initially improves performance, but beyond a certain point, additional compute yields diminishing or no gains; (3) The compute allocated to earlier subtasks influences the scaling dynamics and compute needs of downstream subtasks. Guided by these insights, we propose **AgentTTS**, a novel LLM-agent-based framework designed to efficiently search for compute-optimal budget allocations in multi-stage tasks. Recent works [20, 56, 49, 25, 23] have shown that LLM-based agents are effective in planning and searching for hyperparameter optimization. Given the capability of LLMs in understanding and following instructions, AgentTTS integrates our observed test-time scaling insights into the LLM-agent search process. AgentTTS consists of three key components: the `Agent`, `Archive`, and `Environment`. The `Agent` begins by generating an initial set of trials based on Insight 1, which guides model preferences for each subtask. These trials are executed by the `Environment`, which evaluates them on the actual task platform and returns performance feedback. The `Archive` stores a history of generated trials, guidelines, and feedback. In subsequent stages, the `Agent` produces new trials and guidelines informed by Insights 2 and 3. This iterative process continues until a predefined stopping criterion is met. By leveraging LLM-based search, AgentTTS offers two advantages: (i) interpretability, through explicit guideline generation that explains decision rationales; and (ii) robustness, in navigating non-smooth search spaces commonly found in test-time scaling.

Our **main contributions** are: (i) We study a novel problem of test-time compute-optimal scaling for multi-stage complex tasks; (ii) We identify three key insights that uncover fundamental patterns of test-time scaling in multi-stage tasks and motivate the design of **AgentTTS**, an efficient LLM-agent-based framework for searching compute-optimal configurations in this problem setting. and (iii) Comprehensive evaluations on six datasets show that AgentTTS achieves strong search efficiency, transparent interpretability in generating new trials, and robustness to non-smooth search landscapes.

## 2 Related Work

**Test-time Scaling and Compute-optimal Strategy.** *Test-time scaling* (TTS) enhances LLM performance by allocating additional compute during inference [2, 45, 39]. Existing methods fall into two categories: *sequential scaling* [24, 6, 33, 26, 48], which iteratively refines outputs but depends on good initial responses, and *parallel scaling*[2, 55, 34, 35, 9, 33, 19, 45], which generates multiple outputs and aggregates them using reward-based selection (e.g., repeated sampling [2, 55], Best-of-N [35], or tree search [45]). Recent work reduces reliance on reward models by using LLMs as

fusers [13, 17, 30, 3]. Parallel scaling is preferred for complex tasks due to better scalability and broader solution coverage [33]. Hence, we adopt **repeated sampling with fusion**. Research on *test-time compute-optimal scaling* shows that small models with optimal strategies can outperform larger ones [2, 45, 19, 54, 33, 41, 40]. Approaches include difficulty-aware model selection [33], reward-guided voting [45], and budget-aware prompting [54]. However, they focus on single-stage tasks, while we extend this to multi-stage tasks, where budgets must be adaptively allocated across interdependent subtasks to maximize overall performance. A more detailed discussion is given in Appendix A.10.1.

**LLMs for Hyperparameter Optimization.** LLMs have become powerful tools for hyperparameter optimization (HPO), surpassing traditional AutoML techniques such as Bayesian Optimization (BO) [15, 31] by leveraging contextual reasoning and prior knowledge [8]. LLM-based HPO research generally follows two directions: (1) reducing the search space and (2) directly generating hyperparameter configurations. For the former, LLMs have been used to prune large search spaces in NAS and HPO [52, 25, 23, 21], as seen in GPT-NAS [52], AutoM3L [23], and Llambo [21]. For the latter, recent works [4, 62, 58, 1, 12, 56, 18, 57] treat LLMs as autonomous optimizers. Systems like AutoMMLab [49], GENIUS [62], MLCopilot [56], and AgentHPO [20] refine trials via feedback and experience. We extend this line of work to compute-optimal test-time scaling in multi-stage tasks. A more detailed introduction of related work is given in Appendix A.10.2.

# 3   Preliminary Knowledge and Problem Definition

**Problem Definition**. We define a multi-stage complex task $\mathcal{T} = [T_1, T_2, \ldots, T_n]$ as comprising $n$ simpler subtasks. Each subtask $T_i$ has a set of candidate models $M_i \in \mathcal{M}_i$, where each $M_i$ is tailored for subtask $T_i$. Given a fixed total computational budget $B$ for the entire complex task $\mathcal{T}$, each subtask must be assigned a portion $B_i$ such that $\sum_{i=1}^{n} B_i = B$. For a subtask $T_i$, there exists a trade-off between using a larger model with fewer inference samples and a smaller model with more samples, constrained by the assigned budget $B_i$. Our research problem is defined as

**Definition 1** (Test-time compute-optimal budget allocation in multi-stage tasks). *Given a fixed total computational budget $B$ for a multi-stage complex task $\mathcal{T}$, how can we optimally allocate the computational budget among subtasks $B \mapsto \{B_1, B_2, \ldots, B_n\}$, select appropriate models $M_i$, and effectively distribute the allocated resources to maximize overall performance?*

**Test-time Scaling Mode: Repeated Sampling with Fusion**. We adopt the *Repeated Sampling with Fusion* strategy for test-time scaling, as it does not rely on additional reward models or verifiers compared to Best-of-N [35, 9] or tree search algorithms [45, 53], and it offers greater scalability compared to sequential scaling [55, 2]. Given a problem $p$, a language model $M$ with parameters $\theta$, test-time scaling is performed by increasing the number of repeated samples $k$. To stimulate diverse generations, we set the temperature hyperparameter to 0.9 throughout. A fusion prompt is then used to aggregate the multiple candidate solutions generated through repeated sampling:

$$o = f_{\text{fuse}}(\mathcal{S}, M), \quad \mathcal{S} = \{s_i \mid 1 \leq i \leq k\}, \quad s_i \sim M(s \mid p, \theta) \tag{1}$$

where $\mathcal{S}$ denotes the set of sampled responses, each $s_i$ is independently drawn from the model $M$ conditioned on the input prompt $p$ and model parameters $\theta$. The fusion function $f_{\text{fuse}}$ integrates the sampled responses using a fusion prompt, as described in Appendix A.7. Notably, the same LLM is used for both generating solutions and performing fusion.

# 4   Insights of Test-time Scaling on Multi-stage Complex Tasks

Allocating computational budgets for TTS in multi-stage complex tasks has significant challenges: (i) the search space expands exponentially as the number of subtasks increases, making exhaustive search impractical; (ii) subtasks typically require models tailored to their specific characteristics, rendering uniform model assignment inefficient; (iii) the scaling strategies employed in earlier subtasks directly impact subsequent stages. To address the challenges, we first conduct pilot experiments to understand fundamental patterns in LLM behavior under TTS in complex tasks, which pave the way to design a compute-optimal scaling strategy specifically tailored for multi-stage scenarios.

## 4.1 Experimental Setting

In this subsection, we first briefly introduce the datasets, models, and metrics used across the four multi-stage complex tasks. Then, we describe a unified budget conversion approach across models and tasks, followed by an example using the inference FLOPs as the primary compute cost metric. Detailed descriptions of the datasets, models, and metrics are provided in Appendix A.4.

**Tasks, Datasets, and Models** We conduct preliminary experiments on four multi-stage tasks: (i) Retrieval-based Question Answering using 2WikiMultiHopQA [11] and HotpotQA [50] datasets, (ii) Knowledge Graph Question Answering using CWQ [36] and WebQSP [51], (iii) Task Automation, using TaskBench [32], and (iv) Automated Software Development, using ChatDev [29]. Subtasks differ in prompt and generation lengths; retrieval uses longer prompts, while QA needs longer generations. We consider models ranging from 3B to 72B. Details of task specifications, datasets, models, and evaluation metrics are provided in Appendix A.4 and A.5.

**Unified Budget Conversion Across Models and Tasks** Different models and tasks demand varying levels of computational resources. Larger models require more inference FLOPs, and complex tasks typically involve longer input or output tokens, leading to increased compute costs. These disparities complicate fair comparisons of computational overhead across models and tasks. To address this, we propose a budget normalization framework that equates the cost of fewer inference samples from larger models with that of more samples from smaller models, while explicitly accounting for task-specific compute variations.

Formally, let the total computational cost of sampling $S$ times from model $M$ on task $T$ be denoted as $f_{\text{cost}}(M, S, T)$, and the corresponding normalized compute budget as $B = f_{\text{budget}}(M, S, T)$. The cost metric may reflect user-defined preferences, such as inference FLOPs, wall-clock time, or monetary cost. To establish a common **unit of budget**, we define it as the cost of a single inference pass by the smallest model (e.g., LLaMA 3B) on the lowest computationally consuming task $T_{\text{lowest}}$:

$$f_{\text{budget}}(M_{\text{smallest}}, 1, T_{\text{lowest}}) = 1. \tag{2}$$

Given a model $M_\ell$, sample count $S_\ell$, and task $T_\ell$, we compute the equivalent budget $B$ by equating its total cost to that of the smallest model executing $S_{\text{smallest}}$ samples on the lowest-consuming task:

$$B = f_{\text{budget}}(M_\ell, S_\ell, T_\ell) = S_{\text{smallest}} \text{ such that } f_{\text{cost}}(M_\ell, S_\ell, T_\ell) = f_{\text{cost}}(M_{\text{smallest}}, S_{\text{smallest}}, T_{\text{lowest}}) \tag{3}$$

This formulation expresses the inference budget of any model-task pair as the number of equivalent samples that the smallest model would generate on the least computationally intensive task.

**Compute Cost Metric and Budget Definition.** Different stakeholders prioritize different cost metrics. Consumers are typically concerned with monetary expenses, such as the price per million input and output tokens. LLM researchers focus on computational cost, such as inference FLOPs, due to constraints of limited GPU memory. Discussion of using API price as a cost metric is in Appendix 6. In this work, we adopt inference FLOPs as the primary cost metric throughout. The corresponding budget conversion is stated in the theory below. The proof is in Appendix A.2.

**Theorem 1** (Normalized Budget Function). *Given a configuration $(M_\ell, S_\ell, T_\ell)$ where $M_\ell$ is the model size, $S_\ell$ the number of samples, and $T_\ell = (N_{p,\ell}, N_{d,\ell})$ the task specification where $N_{p,\ell}$ and $N_{d,\ell}$ is the average prompt and generation lengths, the equivalent normalized budget under the base configuration $(M_{smallest} = 3B, N_{p,lowest} = 128, N_{d,lowest} = 64)$ is:*

$$B = f_{budget}(M_\ell, S_\ell, T_\ell) = \frac{2\alpha\beta_2 S_\ell}{\beta_1} + 2(\alpha\beta_2 - 1) \tag{4}$$

*where $\alpha = \frac{M_\ell}{M_{smallest}}, \beta_1 = \frac{N_{p,\ell}}{N_{d,\ell}}, \beta_2 = \frac{N_{p,\ell}}{N_{p,lowest}}$.*

Under this conversion, the budget $B$ intuitively represents: *how many passes using a 3B-parameter model on the base configuration $T_{lowest}$ would incur the same cost as given sampling $S_\ell$ times with model $M_\ell$ on task $T_\ell$?* The base configuration corresponds to the minimal compute setting among our subtasks (see Table 4 in the Appendix) and aligns with standard defaults used in short-form QA tasks, such as QA in WebQSP.

## 4.2 Preliminary Experimental Results and Insights

We use the FLOPs-based budget conversion in Eq. 4 to analyze performance variance with increasing sample count and compute budget, and to guide budget allocation in our method in the next section.

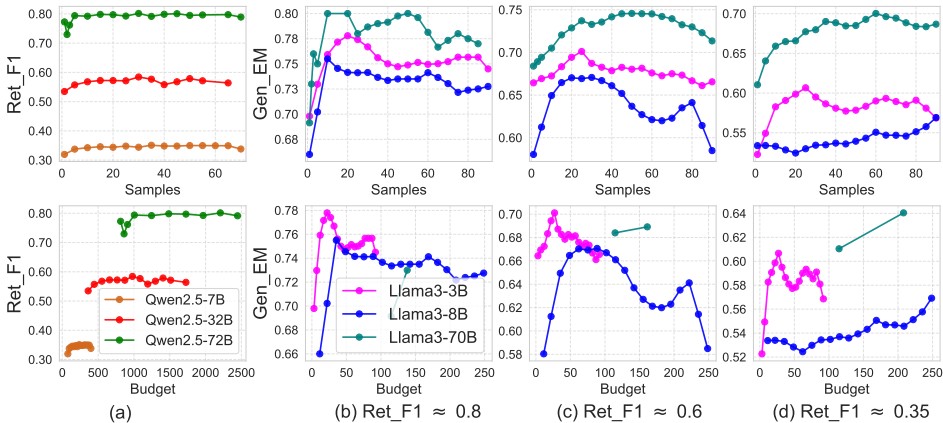

Figure 1: Performance variance on 2WikiMultiHopQA with increasing sampling and inference FLOPs. Top: performance by sample count; bottom: performance by log-scaled inference FLOPs. (a) Retrieval accuracy measured by Retrieval F1 (Ret_F1). (b-d) QA performance under varying retrieval quality levels measured by Gen_EM, the exact match between the generated answer and ground truth

Fig. 1 shows how subtask performance in retrieval-based question answering varies with increasing test-time sampling and computational budget (FLOPs). In Fig. 1 (a), performance on the retrieval subtask exhibits a strong positive correlation with model size, while smaller models provide only marginal improvements even with increased budget. This suggests that retrieval benefits from larger models, where fewer high-capacity samples yield better results. In contrast, Fig. 1 (b) shows that for the question answering subtask, under a limited budget (e.g., $< 10^{14}$ FLOPs), LLaMA-3 3B and LLaMA-3 8B outperform LLaMA-3 70B, indicating a preference for smaller models. Similar trends are observed in Fig. 10 (a)(b) and across other benchmarks (see Fig. 11-14 in Appendix A.6). These discrepancies arise from subtask-specific demands: retrieval emphasizes long-context understanding, favoring large models; while question answering primarily involves extracting information from retrieved content, where smaller models excel and can better exploit test-time scaling via repeated sampling. Thus, **different subtasks exhibit distinct preferences between language models and sampling frequency, depending on the specific capabilities required by each subtask (Insight 1)**.

Fig. 1 (b-d) reveal a non-monotonic performance trend across models in the question answering subtask, which is sensitive to test-time compute. As the number of sampling repetitions increases, performance initially improves but often fluctuates or declines after reaching an optimal budget. For example, LLaMA-3 3B reaches peak performance at 10 samples (FLOPs budget $\approx 2 \times 10^{13}$) in Fig. 9 (b), indicating that additional compute does not always yield better results in this subtask. Similar patterns are observed in Fig. 10 (b-d) and other benchmarks (see Appendix A.6). This is because, as the number of candidates grows, fusion becomes more complex and may become a performance bottleneck. Smaller models, with limited capacity, struggle more under high sampling and tend to degrade, whereas larger models are more capable of handling extensive fusion and show greater tolerance. In conclusion, **subtask-level test-time scaling typically exhibits an optimal budget, beyond which more budget often leads to limited or even negative returns (Insight 2)**.

Fig. 1 (b-d) show how the performance of the question answering subtask varies under different levels of retrieval quality, with F1 scores approximately 0.80, 0.60, and 0.35, respectively. When high-quality retrieval is provided by a larger model (Fig. 1 (b)), we observe: (1) the optimal test-time budget for question answering is reached earlier. For instance, LLaMA-3 8B peaks at 10 samples (budget around $2 \times 10^{13}$ FLOPs); and (2) smaller models (3B and 8B) may outperform the larger model (70B) under the same compute budget. In contrast, when retrieval quality is low (Fig. 1 (c)(d)), the peak performance is delayed. For example, with Ret_F1 = 0.6, LLaMA-3 8B peaks at 20 samples (budget about $3 \times 10^{13}$ FLOPs), while with Ret_F1 = 0.35, even 90 samples (budget near $10^{14}$ FLOPs) still do not reach peak performance. Moreover, under low retrieval quality, smaller models are less likely to outperform larger ones. For instance, in Fig. 1 (d), the peak performance of the 3B and 8B models does not surpass even the lowest performance of the 70B model. These findings indicate that the scaling behavior of a subtask is affected by the performance of its preceding subtasks. Poor retrieval increases downstream task difficulty, requiring the model to compensate for missing information, where larger models are more beneficial. Similar trends are observed in Fig. 10 (b-d) and

**Algorithm 1** Compute-Optimal Test-time Budget Allocation

---
**Require:** Agent $\mathcal{A}$, Model set $\mathcal{M}$, Subtasks $\mathcal{T}$, Environment $\mathcal{E}$, Total budget $B$.
 1: Initialize experiment archive: $\mathcal{L} \leftarrow \emptyset$.
 2: Initialize candidate trials: $\mathcal{C} \leftarrow \mathcal{A}.\texttt{initialize}(\mathcal{M}, \mathcal{T})$                   (Insight 1, Eq. 5)
 3: Obtain feedback: $\mathcal{S} \leftarrow \mathcal{E}.\texttt{execute}(\mathcal{C})$
 4: **while** stopping criterion is not met **do**
 5:      Generate exploration guidelines: $\mathcal{G} \leftarrow \mathcal{A}.\texttt{generate\_guidelines}(\mathcal{C}, \mathcal{S})$      (Insights 2, 3)
 6:      Update experiment log: $\mathcal{L} \leftarrow \mathcal{L} \cup \{(\mathcal{C}, \mathcal{S}, \mathcal{G})\}$
 7:      Generate new candidate trials: $\mathcal{C} \leftarrow \mathcal{A}.\texttt{generate}(\mathcal{G}, \mathcal{M}, \mathcal{T}, B)$              (Eq. 5)
 8:      Obtain feedback: $\mathcal{S} \leftarrow \mathcal{E}.\texttt{execute}(\mathcal{C})$
 9: **end while**
10: **return** Best-performing trial from $\mathcal{L}$.

---

other benchmarks in Appendix A.6. We conclude that **budget allocation for a preceding subtask impacts the model selection and optimal budget of subsequent subtasks (Insight 3)**.

## 5 AgentTTS: Agent for Test-time Scaling Budget Allocation

The insights in Sec. 4 provide guidance on how to efficiently search for effective budget and model allocations in multi-stage tasks. As LLMs have shown strong capabilities in planning and reasoning, we propose **AgentTTS** (**Agent** for **T**est-**T**ime compute-optimal **S**caling), which integrates these insights into an LLM-based agent to autonomously navigate the compute allocation space and capture subtask-specific model preferences and optimal budget, and inter-subtask dependencies. An illustration of AgentTTS is in Fig. 2. Next, we elaborate on the design of AgentTTS.

**Overview of AgentTTS**. As shown in Fig. 2 (b-d), the framework consists of three core components: the `Agent`, `Archive`, and `Environment`. Initially, the `Agent` generates a batch of candidate trials (each looks like $(M_1, B_1, M_2, B_2, \ldots)$), guided by Insight 1, which reflects preliminary model preferences across subtasks. These trials are stored in the `Archive` and forwarded to the `Environment` for evaluation. The resulting performance feedback is returned to the `Agent` and used to construct initial *guidelines* that suggest whether subsequent trials should prioritize smaller or larger models. Both feedback and guidelines are retained in the `Archive` for future reference. In subsequent iterations, the `Agent` generates new trials based on the existing guidelines and adds new guidelines by Insight 2 and Insight 3. The

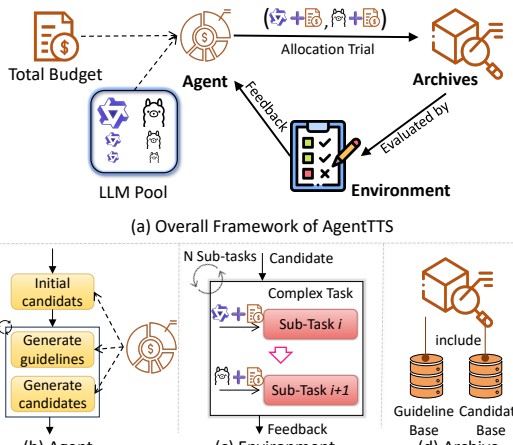

(a) Overall Framework of AgentTTS

(b) Agent     (c) Environment     (d) Archive

Figure 2: Overview of LLM Agent for Test-time Scaling Budget Allocation.

`Archive` continuously stores the evolving guidelines along with corresponding trials and performance feedback, while the `Environment` evaluates each batch of trials in a practical execution environment and returns feedback. This loop repeats until a predefined stopping criterion is met. The full procedure is outlined in Algo. 1, and the class diagram is shown in Fig. 16 (Appendix).

**Agent component** The `Agent`, implemented using an LLM, is responsible for generating test-time budget allocation candidate trials and guidelines, as shown in Fig. 2 (b). Although LLMs perform well in hyperparameter optimization for machine learning models, they lack knowledge of test-time scaling, a relatively new technique. To address this, we incorporate Insight 1 into the initial search stage and Insights 2 and 3 into all stages.

Insight 1 reveals that different subtasks prefer models with specific capabilities. Therefore, matching a suitable model early steers the search toward effective configurations, reducing wasted effort on inferior options. To estimate each subtask's model preference, we perform the following during the initialization stage. Let $B_i^{\max} = f_{\text{budget}}(M_{i,\max}, 1, T_i)$ be the budget required for a single sample from the largest available model $M_{i,\max}$ for subtask $T_i$ and $B_i^{\min} = f_{\text{budget}}(M_{i,\min}, 1, T_i)$ the budget

from the smallest model. For each subtask $\mathcal{T}_i$, we compare all candidate models that fit within $B_i^{\mathrm{max}}$, while fixing all other subtasks to use their respective largest available model with one-pass inference. This ensures a fair comparison across model candidates for the target subtask $\mathcal{T}_i$. Under the condition that $B_i^{\mathrm{max}} > B - \sum_{j \neq i} B_j^{\mathrm{min}}$, the largest model for subtask $i$ occupies too much budget to allow feasible allocations for the remaining subtasks. In this case, we progressively downsize the model until the largest available model $M_{i,\mathrm{max}}$ is found. This operation is repeated for each subtask.

Upon receiving initial feedback from the `Environment` module, the `Agent` summarizes the initial guidelines $\mathcal{G}$ (Algo. Line 5) using the prompt provided in Appendix A.7, based on performance comparisons. If the large model significantly outperforms the small model, the large one is preferred. Otherwise, the small model is preferred due to its greater flexibility in subsequent exploration. The resulting guideline specifies which model should be prioritized in the later stages of the search. The instruction to identify the preferred model is used only in the initial search stage. The overall procedure is implemented in Line 2 of Alg. 1 and formalized in Eq. 5 below

$$
\mathcal{C} = \begin{cases} \{(..., M_i, B_i, ...) \mid 1 \leq i \leq N, \ M_i \in \mathcal{M}_i, \ B_i = B_i^{\mathrm{max}}\}, & \text{initial stage} \\ \{c \mid c \in \mathcal{A}.\texttt{generate}(\mathcal{G}, \mathcal{M}, \mathcal{T}, B)\}, & \text{subsequent stages} \end{cases} \tag{5}
$$

where each trial is represented as $c = (M_1, B_1, \ldots, M_n, B_n)$, $M_i \in \mathcal{M}_i$ is the selected model, and $B_i$ is its assigned budget for subtask $i$. $B$ denotes the total compute budget. $M_{i,\mathrm{max}}$ is the largest available model for subtask $i$, and the configuration $(..., M_i, B_i, ...)$ indicates that other subtasks are fixed to their respective available largest models with one-pass inference.

Insight 2 shows that increasing test-time compute initially improves performance, but beyond an optimal point, it may cause oscillation or degradation within a subtask. Each model thus has a task-specific optimal budget: too little limits performance, while too much wastes compute and harms other subtasks. Identifying this optimal budget per subtask is key to maximizing overall performance efficiently. To support this, we put Insight 2 in the guideline generation prompt (Appendix A.7) and ask LLM to follow Insight 2 to: "identify the search direction for finding the optimal number of samples for each subtask." This ensures that the agent focuses the next round of search within the right scope, enabling faster convergence to optimal allocation.

Insight 3 indicates that budget allocation for a preceding subtask affects model choice and sampling needs in downstream subtasks. Under limited budgets, optimal configurations cannot be assigned to all subtasks. Allocating more resources to one may shift the optimal setup of others, making previous configurations suboptimal. This interdependence increases search complexity, as each subtask must be re-evaluated in light of others' changes. To address this, we embed an instruction into the prompt (Appendix A.7) that leverages the LLM's planning capabilities to explore allocation trade-offs across subtasks. This enables the `Agent` to generate search guidelines that adaptively identify critical subtasks and configurations.

The instructions derived from three insights are applied concurrently throughout the search process. Based on the updated guidelines, the `Agent` then generates the next round of candidate trials (Algo. Line 7) for evaluation.

**Environment & Archive Components** The `Environment` module executes and evaluates trials in the actual runtime environment. Upon receiving trials from the `Archive`, it converts them into executable scripts and submits them to the task platform for execution on a small training set. After completion, performance feedback is returned to the `Agent`, as shown in Fig. 2(c) and Lines 3 and 8 of Algo. 1. The `Archive` component stores generated guidelines and candidate trials in corresponding bases, as shown in Fig. 2(d) and Lines 1 and 6 of Algo. 1. It records the search process throughout iterations and outputs the best-performing trials upon termination.

## 6 Experiments

This section presents a comprehensive evaluation of **AgentTTS** for test-time compute-optimal budget allocation in multi-stage complex tasks. We also conduct ablation studies on integrated key insights, analyze the interpretability of budget allocations, evaluate robustness to different training sizes, and compare search effectiveness under various budget conversion methods.

**Experimental Setup**. We conduct experiments across six datasets covering four distinct task categories, as detailed in Appendix A.4. Baselines include two groups of hyperparameter optimization:

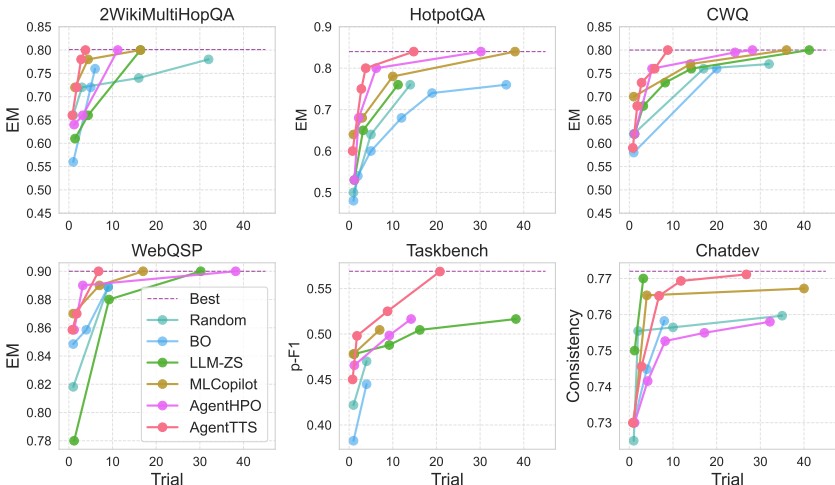

Figure 3: Performance trajectories across various search methods over 50 trials. X-axis: trial count; Y-axis: best performance up to each trial. The best score is obtained from the optimal trial in a prior grid search and serves as the benchmark for all methods.

traditional machine learning methods and recent LLM-based methods. The traditional baselines include Bayesian optimization (BO) [15, 31] and random search. For LLM-based approaches, we consider AgentHPO [20], MLCopilot [56], and LLM_ZS. Given that these LLM-based methods were initially proposed for hyperparameter tuning, we adapt them to the test-time budget allocation. Both AgentHPO and MLCopilot employ a two-phase pipeline: (1) feedback-driven guideline generation and (2) guideline-informed candidate trial generation. They differ in the initialization of guidelines: AgentHPO generates initial guidelines autonomously via the LLM, while MLCopilot initializes guidelines based on performance trends observed in similar tasks. In contrast, LLM_ZS directly prompts the LLM to generate candidate trials without relying on guidelines or external references. Further details of each baseline are in Appendix A.9. All methods execute 50 iterations of search on a training set comprising 50 samples and are subsequently evaluated on a test set containing 500 samples. We use GPT-o3-mini [27] as the LLM search agent. The default budget is set as the sum of the minimum budget required for one-pass inference on each subtask using the largest model.

**Main Results and Analysis**. Fig. 3 shows the search trajectories of our method and the baselines for test-time compute-optimal budget allocation on the training set. The search time and test-set performance of the best trials are reported in Tab. 1. We observe: (i) Our method outperforms baselines in both search efficiency and test-set performance in Fig. 3 and Tab. 1. First, ours achieves top results on most tasks and requires fewer trials, even when baselines eventually reach optimality. Second, our method reduces search time, confirming higher computational efficiency. These gains come from leveraging empirical insights to guide the search toward promising trials. Despite similar performance on the training set, our method often generalizes better, as shown in Tab. 1. For example, on 2WikiMultiHopQA, it exceeds the next best methods by 2% on the test set. This suggests that Insight 2 helps identify minimal budgets and avoid redundant sampling, improving generalization. (ii) Compared to LLM-agent-based AgentHPO and MLCopilot, our method converges faster, showing integrated insights improve search efficiency on compute-optimal allocation. (iii) Traditional methods such as BO often get stuck in local optima due to the non-smooth landscape, while random search is noise-tolerant but inefficient. In contrast, LLM-based methods use prior hyperparameter tuning knowledge to bypass suboptimal regions, achieving better search performance.

Table 1: Comparison of search time (in hours) and test-set performance across datasets. "–" indicates unavailable results or failure to find the optimal trial.

| Method | 2Wiki | | Hotpot | | CWQ | | WebQSP | | Taskbench | | ChatDev | |
|---|---|---|---|---|---|---|---|---|---|---|---|---|
| | Time | EM | Time | EM | Time | EM | Time | EM | Time | p-F1 | Time | Cons. |
| Random | – | 0.66 | – | 0.71 | – | 0.76 | – | 0.86 | – | 0.40 | – | 0.74 |
| BO | – | 0.60 | – | 0.71 | – | 0.76 | – | 0.85 | – | 0.52 | – | 0.75 |
| LLM_ZS | 12.5 | 0.70 | – | 0.71 | 55.3 | 0.76 | 37.7 | 0.89 | – | 0.49 | – | 0.74 |
| MLCopilot | 12.5 | 0.70 | 46.3 | 0.72 | 48.4 | 0.78 | 20.8 | 0.88 | – | 0.53 | – | 0.75 |
| AgentHPO | 8.3 | 0.70 | 36.3 | 0.74 | 37.4 | 0.78 | 48.1 | 0.89 | – | 0.49 | – | 0.74 |
| Ours | 2.5 | 0.72 | 17.5 | 0.74 | 11.1 | 0.78 | 7.8 | 0.89 | 64.3 | 0.53 | 14.3 | 0.75 |

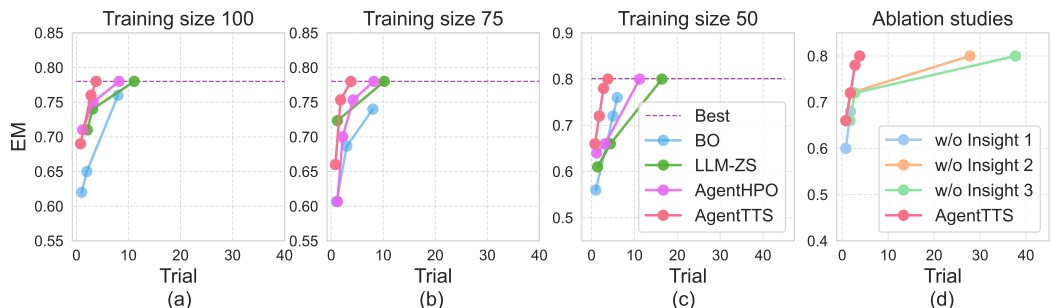

Figure 4: (a-c) Search trajectories under varying training sizes on 2WikiMultiHopQA. (d) Ablation study on 2WikiMultiHopQA with a total budget of 900.

**Ablation Studies**. To evaluate the contribution of each insight in AgentTTS, we perform ablation studies by comparing it to variants with individual insights removed. AgentTTS-w/o-Insight1 replaces Insight 1 with random initialization. AgentTTS-w/o-Insight2/3 removes prompt components addressing diminishing returns beyond optimal budget (Insight 2) and inter-subtask dependencies (Insight 3). As shown in Fig. 4(d), we observe: (1) Removing Insight 1 prevents reaching optimal configurations, highlighting the role of initial model choice in guiding subsequent exploration. (2) Without Insight 2, search efficiency drops, delaying optimal trials to 29 steps, showing the need to identify per-subtask optimal budget. (3) Excluding Insight 3 delays the optimal trial to step 38, confirming the importance of leveraging LLM planning to handle subtask dependencies.

**Robustness to Varying Training Sizes**. A small training set might lead to unstable performance, creating a non-smooth search landscape that hinders optimization. To assess the robustness of search methods under this condition, we vary the training set size (50, 75, 100 samples) and evaluate their effectiveness in finding optimal configurations. As shown in Fig. 4(a-c), the search efficiency of LLM-based baselines and Bayesian optimization declines with smaller training sets, whereas AgentTTS maintains strong efficiency. This indicates that our empirical insights integrated in AgentTTS help the LLM agent effectively navigate non-smooth search spaces through valuable prior guidance.

**Interpretability of Budget Allocation** To illustrate the interpretability benefit from integrating prior insights into the LLM agent for test-time compute allocation, we present three case studies on 2WikiMultiHopQA in Fig. 5. The first row shows Insight 1 enables subtask-specific model use, favoring large models for retrieval and small ones for QA, to guide efficient trials. The second row shows Insight 2 pinpoints the optimal sampling range (5–50), narrowing the search space. The third row shows Insight 3 promotes balanced allocation, favoring one-sample "high-capacity" retrieval and "lower-cost" QA configuration, enabling effective trade-offs for better overall performance.

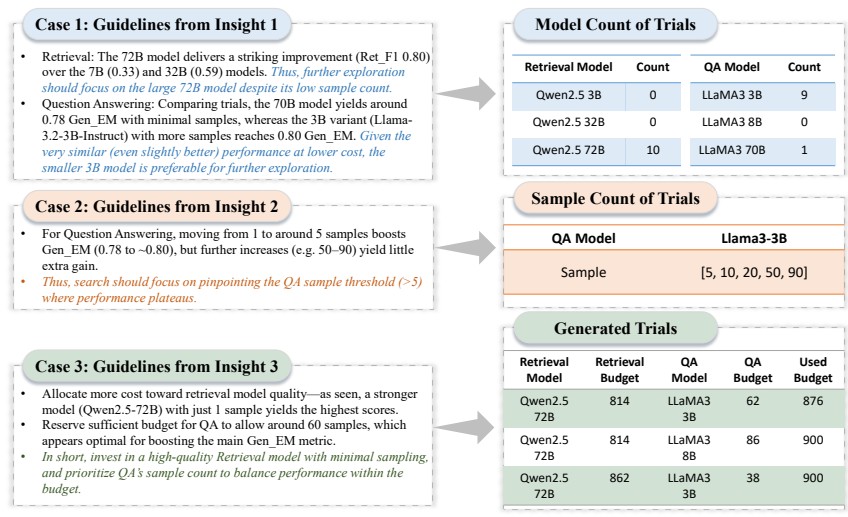

Figure 5: Detailed Cases of Interpreting AgentTTS Through Integrated Empirical Insights.

**Performance under varying budget settings**
To assess the search efficiency under different budgets, we evaluate AgentTTS under two budget settings on 2WikiMultiHopQA: 500 (low budget: only one subtask can reach its optimum) and 2000 (high, full optima reachable but with larger search spaces). As shown in Fig. 6, we can conclude: (1) Our method finds optimal configurations under low budgets, confirming the benefit of integrated insights on trade-off subtask interdependence; (2) Across low to high budget settings, our method consistently outperforms baselines in search efficiency, indicating that the insight-informed LLM agent remains robust to the expanded search space.

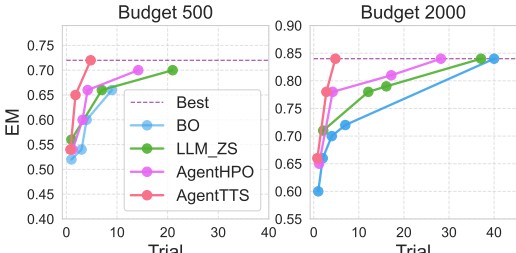

Figure 6: Comparative search results under low, medium, and high compute budget settings.

**Performance under varying temperatures** To assess the impact of temperature on test-time scaling, we conduct additional experiments on the 2Wiki dataset. The retriever is fixed to Qwen2.5-72B (producing a single retrieval result), and we vary the number of samples and temperature values for the QA model LLaMA 3 3B. As shown in Tab. 2, higher temperatures (e.g., 0.9) improve performance with multiple samples, while lower temperatures (e.g., 0.1) are more effective for single-sample cases. This suggests that higher temperatures enhance output diversity and fusion in sampling-based reasoning, whereas lower temperatures promote more stable, deterministic outputs. These results align with prior findings [28], which show that temperature-induced diversity improves sample efficiency in reasoning tasks.

**API price as the cost metric** Beyond inference-compute FLOPs, end-users are often more concerned with the monetary cost. Therefore, we introduce an alternative budget metric: API price. Using this metric, we redraw the test-time scaling curves for the question answering subtask of 2WikiMultiHopQA, as shown in Fig. 7 (left). The results indicate that smaller models remain preferable under constrained budgets. Furthermore, as illustrated in Fig. 7 (right), the proposed insights continue to improve the search efficiency and performance of AgentTTS, demonstrating strong generalization across different cost metrics.

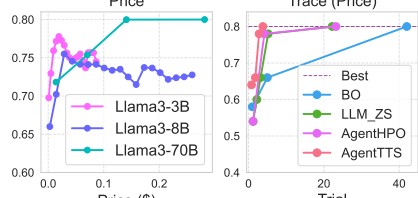

Figure 7: Test-time scaling and AgentTTS search trajectories under price budget. Left: scaling curves; Right: corresponding search trajectories.

## 7 Conclusion

We study a novel problem of test-time compute-optimal scaling in multi-stage complex tasks, where the search space grows exponentially with the number of model choices and subtasks, and budget allocation across subtasks is interdependent. Empirical analysis across four task types and six datasets yields three key insights: (1) subtasks have distinct preferences for small or large models; (2) test-time scaling saturates, with diminishing or negative returns beyond an optimal budget; and (3) early subtask budgets influence downstream scaling behavior. Informed by these insights, we propose AgentTTS, an LLM-agent-based framework that iteratively

Table 2: EM scores of LLaMA 3 3B at different temperature values on the 2Wiki dataset. The retriever is fixed to Qwen2.5-72B (single retrieval result).

| Samples | Temp = 0.1 | Temp = 0.5 | Temp = 0.9 |
|---|---|---|---|
| 1 | 0.70 | 0.64 | 0.66 |
| 10 | 0.70 | 0.78 | 0.78 |
| 20 | 0.72 | 0.76 | 0.78 |
| 30 | 0.74 | 0.78 | 0.72 |
| 40 | 0.74 | 0.76 | 0.78 |
| 50 | 0.74 | 0.72 | 0.78 |
| 60 | 0.74 | 0.78 | 0.80 |
| 70 | 0.76 | 0.72 | 0.74 |
| 80 | 0.76 | 0.76 | 0.78 |
| 90 | 0.74 | 0.72 | 0.76 |

searches for compute-optimal configurations through interaction with actual task platforms. Experiments show that AgentTTS outperforms both traditional and LLM-based baselines in search efficiency, final test performance, and robustness to non-smooth landscapes.

## Acknowledgment

This material is based on work supported by, or in part by, the Army Research Office (ARO) under grant number W911NF-21-1-0198 and Cisco Faculty Research Award.

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

# A  Technical Appendices and Supplementary Material

## A.1  An Example of Search-space Size

We take automated software development as a representative example, which consists of three subtasks: coding, static testing, and dynamic testing. The average prompt and generation token lengths for each subtask are summarized in Table 4. All subtasks share the same model space $\mathcal{M} = \{\text{LLaMA-3 3B}, \text{LLaMA-3 70B}\}$. Given a total compute budget of $B = \sum_i B_i^{\max} \approx 1768$, which corresponds to the cost of using the largest model (LLaMA-3 70B) for all subtasks, we provide the corresponding calculation code below. The output shows the count is about $1.8 \times 10^6$.

```python
def compute_budget(S, M_large, Np1, Nd1, M_small=3e9, Np2=128, Nd2=64):
    '''
    Compute the normalized compute budget required to generate S samples from a model
        M_large on a task with prompt length Np1 and generation length Nd1.

    The budget is normalized with respect to the unit cost of generating 1 sample using
        the smallest model (M_small) on the lowest computationally consuming task (
        prompt_length=Np2, decoding_length=Nd2).
    '''
    alpha = M_large / M_small  # Model size ratio
    beta1 = Np1 / Nd1  # Prompt-to-generation length ratio for the target task
    beta2 = Np1 / Np2  # Prompt length scaling factor relative to the reference task
    beta3 = Np2 / Nd2  # Prompt-to-generation length ratio for the reference task
    # Compute normalized budget cost
    return beta3 * ((alpha * beta2 / beta1) * (beta1 + S) - 1)

def find_sample_upper_bound(max_budget, model_size, prompt_len, decode_len):
    '''Iteratively increases S (sample count) until budget is exceeded for a given model
        and task configuration.'''
    for S in range(1, 10000):
        if compute_budget(S, model_size, prompt_len, decode_len) > max_budget:
            return S - 1
    return 9999

# Compute the total budget required for one sample from the largest model across tasks.
b = compute_budget(1, 70e9, 1024, 1024) + compute_budget(1, 70e9, 1024, 512) +
    compute_budget(1, 70e9, 1024, 256)
model_space = [3e9, 70e9]  # Two model sizes: small (3B) and large (70B)
# Compute the maximum feasible sample count per task using the smallest model
max_s1 = find_sample_upper_bound(b, 3e9, 1024, 1024)
max_s2 = find_sample_upper_bound(b, 3e9, 1024, 512)
max_s3 = find_sample_upper_bound(b, 3e9, 1024, 256)
Np1_list = [1024, 1024, 1024]  # Prompt lengths for each subtask
Nd1_list = [1024, 512, 256]  # Generation lengths for each subtask

# Exhaustive search over valid configurations under the total budget constraint
valid_config_count = 0
for s1 in range(max_s1 + 1):
    for model1 in model_space:
        b1 = compute_budget(s1, model1, Np1_list[0], Nd1_list[0])
        if b1 > b: break

        for s2 in range(max_s2 + 1):
            for model2 in model_space:
                b2 = compute_budget(s2, model2, Np1_list[1], Nd1_list[1])
                if b1 + b2 > b: break

                for s3 in range(max_s3 + 1):
                    for model3 in model_space:
                        b3 = compute_budget(s3, model3, Np1_list[2], Nd1_list[2])
                        if b1 + b2 + b3 > b: break
                        valid_config_count += 1

print("Total valid configurations:", valid_config_count)
# The output should be: Total valid configurations: 1854841
```

Listing 1: Enumerating Valid (M1, S1, M2, S2, M3, S3) Triplets under Budget Constraint

## A.2  Inference FLOPs-Based Budget Conversion

We adopt floating-point operations (FLOPs) as the primary computation cost metric for measuring inference cost. In repeated sampling scenarios, Transformer models can exploit batching to share

the cost of prompt encoding across multiple decoding passes. This allows for efficient reuse of the prompt representation while performing multiple, independent decoding operations.

Then, we formalize the computation of FLOPs and describe how to convert the number of decoding samples between a large and a small model under an equivalent FLOPs budget. We use unified notation throughout in Tab. 3.

Table 3: Notation used in FLOPs-based cost formulation.

| Symbol | Description |
|---|---|
| $M$ | Number of model parameters |
| $L$ | Number of Transformer layers |
| $D$ | Hidden dimension size |
| $N_p$ | Number of prompt tokens |
| $N_d$ | Number of generated (decoded) tokens per sample |
| $S$ | Number of decoding samples |

The calculation of budget conversion using inference FLOPs in Eq. 4 is as follows:

*Proof.* **Per-token FLOPs.** For a decoder-only Transformer generating tokens autoregressively, the per-token FLOPs primarily stem from two components:

- **(i) Parameter matrix multiplications.** Each non-embedding weight is used exactly once per token in a matrix multiplication followed by addition, resulting in approximately $2M$ FLOPs per token, where $M$ is the total number of non-embedding parameters.

- **(ii) Attention operations.** At decoding step $t + 1$, attention requires computing query-key dot products and applying the attention weights to value vectors. Both operations involve multiply-adds, contributing approximately $4LDt$ FLOPs per token, where $L$ is the number of layers, $D$ is the hidden size, and $t$ is the number of preceding tokens.

Thus, the approximate FLOPs per token is: $\text{FLOPs}_{\text{token}}(M, L, D, t) \approx 2M + 4LDt$.

**Per-phase FLOPs.** The total inference cost for a generation can be decomposed into two phases:

*(1) Prompt Encoding:*

$$\text{FLOPs}_{\text{prompt}}(M, L, D, N_p) = \sum_{t=1}^{N_p} \text{FLOPs}_{\text{token}}(M, L, D, t)$$
$$= 2MN_p + 2LDN_p(N_p + 1) \tag{6}$$

*(2) Decoding:*

$$\text{FLOPs}_{\text{decode}}(M, L, D, N_p, N_d) = \sum_{t=1}^{N_d} \text{FLOPs}_{\text{token}}(M, L, D, N_p + t)$$
$$= 2MN_d + 2LDN_d(2N_p + N_d + 1) \tag{7}$$

*(3) Total FLOPs:* To generate $S$ decoding samples from a prompt of length $N_p$, the total FLOPs is:

$$\text{FLOPs}_{\text{total}}(S, M, L, D, N_p, N_d) = \text{FLOPs}_{\text{prompt}}(M, L, D, N_p) + S \cdot \text{FLOPs}_{\text{decode}}(M, L, D, N_p, N_d) \tag{8}$$

We denote this as $f_{\text{cost}} = \text{FLOPs}_{\text{total}}(S, M, L, D, N_p, N_d)$.

**Budget Conversion Across Models and Tasks.** Consider converting FLOPs usage between two tasks using different models: a large model ($\ell$) for Task 1 and a smaller model ($s$) for Task 2. Let $M_\ell, L_\ell, D_\ell$ and $M_s, L_s, D_s$ be their respective parameters, and $N_{p,1}, N_{d,1}$, and $N_{p,2}, N_{d,2}$ the average prompt and decoding lengths for the two tasks.

Suppose the large model executes $S_\ell$ decoding samples. The equivalent number of samples $S_s$ that the small model can generate under the same FLOPs budget is:

$$S_s = \frac{\text{FLOPs}_{\text{total},\ell}(S_\ell, M_\ell, L_\ell, D_\ell, N_{p,1}, N_{d,1}) - \text{FLOPs}_{\text{prompt},s}(M_s, L_s, D_s, N_{p,2})}{\text{FLOPs}_{\text{decode},s}(M_s, L_s, D_s, N_{p,2}, N_{d,2})} \tag{9}$$

To simplify, we ignore attention-related terms (typically less than 1% of total FLOPs) and define: $\beta_1 = \frac{N_{p,1}}{N_{d,1}}, \beta_2 = \frac{N_{p,1}}{N_{p,2}}, \beta_3 = \frac{N_{p,2}}{N_{d,2}}, \alpha = \frac{M_\ell}{M_s}$. Substituting these yields:

$$S_s = \beta_3 \left( \frac{\alpha\beta_2}{\beta_1}(\beta_1 + S_\ell) - 1 \right) \tag{10}$$

**Normalized Budget Function.** We define the unit budget $B = 1$ as the FLOPs cost of generating a single sample using the smallest model ($M_s = 3$B) on the lowest consuming task specification, characterized by $N_{p,\text{lowest}} = 128$ and $N_{d,\text{lowest}} = 64$. This setting represents the minimal computational cost among all available subtask-model combinations.

Substituting into Eq. 10 yields:

$$S_s = \frac{2\alpha\beta_2 S_\ell}{\beta_1} + 2(\alpha\beta_2 - 1) \tag{11}$$

Thus, for a configuration defined by model size $M_\ell$, sample count $S_\ell$, and task parameters $T_\ell = (N_{p,\ell}, N_{d,\ell})$, the normalized budget function is:

$$f_{\text{budget}}(M_\ell, S_\ell, T_\ell) = S_s = \frac{2\alpha\beta_2 S_\ell}{\beta_1} + 2(\alpha\beta_2 - 1) \tag{12}$$

$\square$

We provide the average lengths of prompt and decoding in different subtasks in Tab. 4. We provide the look-up tables when generating $S = 1, 5, 10, 45, 90$ in Tab. 5–9.

Table 4: Average prompt and generation token lengths ($N_p$, $N_d$) for each subtask.

| Subtask | $N_p$ | $N_d$ |
|---|---|---|
| 2WikiMultiHopQA-Retrieval | 2048 | 128 |
| 2WikiMultiHopQA-QA | 256 | 64 |
| HotpotQA-Retrieval | 2048 | 128 |
| HotpotQA-QA | 256 | 64 |
| CWQ-Retrieval | 1024 | 64 |
| CWQ-QA | 256 | 64 |
| WebQSP-Retrieval | 1024 | 64 |
| WebQSP-QA | 128 | 64 |
| Taskbench-Decomposition | 1024 | 64 |
| Taskbench-Tool Selection | 1024 | 256 |
| Taskbench-Parameter Prediction | 1024 | 2048 |
| ChatDev-Code | 1024 | 1024 |
| ChatDev-Static Test | 1024 | 512 |
| ChatDev-Dynamic Test | 1024 | 256 |

Table 5: Normalized compute budget when generating $S = 1$ sample.

| Model | 2Wiki-R | 2Wiki-QA | Hot-R | Hot-QA | CWQ-R | CWQ-QA | WQSP-R | WQSP-QA | Decomp | ToolSel | ParamPred | Code | Static | Dynamic |
|---|---|---|---|---|---|---|---|---|---|---|---|---|---|---|
| Qwen2.5-72B | 814 | 118 | 814 | 118 | 406 | 118 | 406 | 70 | 406 | 478 | 1150 | 766 | 574 | 478 |
| Qwen2.5-32B | 361 | 51 | 361 | 51 | 179 | 51 | 179 | 30 | 179 | 211 | 510 | 339 | 254 | 211 |
| Qwen2.5-7B | 77 | 10 | 77 | 10 | 38 | 10 | 38 | 5 | 38 | 45 | 110 | 73 | 54 | 45 |
| LLaMA-3.1-70B | 791 | 115 | 791 | 115 | 395 | 115 | 395 | 68 | 395 | 465 | 1118 | 745 | 558 | 465 |
| LLaMA-3.1-8B | 89 | 11 | 89 | 11 | 43 | 11 | 43 | 6 | 43 | 51 | 126 | 83 | 62 | 51 |
| LLaMA-3.2-3B | 32 | 3 | 32 | 3 | 15 | 3 | 15 | 2 | 15 | 18 | 46 | 30 | 22 | 18 |
| Gemma-2-27B | 304 | 43 | 304 | 43 | 151 | 43 | 151 | 26 | 151 | 178 | 430 | 286 | 214 | 178 |
| Gemma-2-9B | 100 | 13 | 100 | 13 | 49 | 13 | 49 | 8 | 49 | 58 | 142 | 94 | 70 | 58 |
| Gemma-2-2B | 21 | 1 | 21 | 1 | 9 | 1 | 9 | 1 | 9 | 11 | 30 | 19 | 14 | 11 |
| Phi-3-medium | 157 | 21 | 157 | 21 | 77 | 21 | 77 | 13 | 77 | 91 | 222 | 147 | 110 | 91 |
| Phi-3-small | 77 | 10 | 77 | 10 | 38 | 10 | 38 | 5 | 38 | 45 | 110 | 73 | 54 | 45 |
| Phi-3-mini | 41 | 4 | 41 | 4 | 20 | 4 | 20 | 3 | 20 | 23 | 59 | 39 | 28 | 23 |

Table 6: Normalized compute budget when generating $S = 5$ samples.

| Model | 2Wiki-R | 2Wiki-QA | Hot-R | Hot-QA | CWQ-R | CWQ-QA | WQSP-R | WQSP-QA | Decomp | ToolSel | ParamPred | Code | Static | Dynamic |
|---|---|---|---|---|---|---|---|---|---|---|---|---|---|---|
| Qwen2.5-72B | 1006 | 214 | 1006 | 214 | 502 | 214 | 502 | 166 | 502 | 862 | 4222 | 2302 | 1342 | 862 |
| Qwen2.5-32B | 446 | 94 | 446 | 94 | 222 | 94 | 222 | 73 | 222 | 382 | 1875 | 1022 | 595 | 382 |
| Qwen2.5-7B | 96 | 19 | 96 | 19 | 47 | 19 | 47 | 14 | 47 | 82 | 409 | 222 | 129 | 82 |
| LLaMA-3.1-70B | 978 | 208 | 978 | 208 | 488 | 208 | 488 | 161 | 488 | 838 | 4105 | 2238 | 1305 | 838 |
| LLaMA-3.1-8B | 110 | 22 | 110 | 22 | 54 | 22 | 54 | 17 | 54 | 94 | 467 | 254 | 147 | 94 |
| LLaMA-3.2-3B | 40 | 7 | 40 | 7 | 19 | 7 | 19 | 6 | 19 | 34 | 174 | 94 | 54 | 34 |
| Gemma-2-27B | 376 | 79 | 376 | 79 | 187 | 79 | 187 | 61 | 187 | 322 | 1582 | 862 | 502 | 322 |
| Gemma-2-9B | 124 | 25 | 124 | 25 | 61 | 25 | 61 | 20 | 61 | 106 | 526 | 286 | 166 | 106 |
| Gemma-2-2B | 26 | 4 | 26 | 4 | 12 | 4 | 12 | 3 | 12 | 22 | 115 | 62 | 35 | 22 |
| Phi-3-medium | 194 | 40 | 194 | 40 | 96 | 40 | 96 | 31 | 96 | 166 | 819 | 446 | 259 | 166 |
| Phi-3-small | 96 | 19 | 96 | 19 | 47 | 19 | 47 | 14 | 47 | 82 | 409 | 222 | 129 | 82 |
| Phi-3-mini | 51 | 9 | 51 | 9 | 25 | 9 | 25 | 8 | 25 | 44 | 221 | 120 | 69 | 44 |

Table 7: Normalized compute budget for each subtask when generating $S = 10$ samples.

| Model | 2Wiki-R | 2Wiki-QA | Hot-R | Hot-QA | CWQ-R | CWQ-QA | WQSP-R | WQSP-QA | Decomp | ToolSel | ParamPred | Code | Static | Dynamic |
|---|---|---|---|---|---|---|---|---|---|---|---|---|---|---|
| Qwen2.5-72B | 1246 | 334 | 1246 | 334 | 622 | 334 | 622 | 214 | 622 | 1342 | 8062 | 4222 | 2302 | 1342 |
| Qwen2.5-32B | 553 | 147 | 553 | 147 | 275 | 147 | 275 | 94 | 275 | 595 | 3582 | 1875 | 1022 | 595 |
| Qwen2.5-7B | 119 | 31 | 119 | 31 | 59 | 31 | 59 | 19 | 59 | 129 | 782 | 409 | 222 | 129 |
| LLaMA-3.1-70B | 1211 | 325 | 1211 | 325 | 605 | 325 | 605 | 201 | 605 | 1305 | 7838 | 4105 | 2238 | 1305 |
| LLaMA-3.1-8B | 137 | 35 | 137 | 35 | 67 | 35 | 67 | 22 | 67 | 147 | 894 | 467 | 254 | 147 |
| LLaMA-3.2-3B | 50 | 12 | 50 | 12 | 24 | 12 | 24 | 8 | 24 | 54 | 334 | 174 | 94 | 54 |
| Gemma-2-27B | 466 | 124 | 466 | 124 | 232 | 124 | 232 | 76 | 232 | 502 | 3022 | 1582 | 862 | 502 |
| Gemma-2-9B | 154 | 40 | 154 | 40 | 76 | 40 | 76 | 25 | 76 | 166 | 1006 | 526 | 286 | 166 |
| Gemma-2-2B | 33 | 7 | 33 | 7 | 15 | 7 | 15 | 5 | 15 | 35 | 222 | 115 | 62 | 35 |
| Phi-3-medium | 241 | 63 | 241 | 63 | 119 | 63 | 119 | 39 | 119 | 259 | 1566 | 819 | 446 | 259 |
| Phi-3-small | 119 | 31 | 119 | 31 | 59 | 31 | 59 | 19 | 59 | 129 | 782 | 409 | 222 | 129 |
| Phi-3-mini | 64 | 16 | 64 | 16 | 31 | 16 | 31 | 10 | 31 | 69 | 424 | 221 | 120 | 69 |

Table 8: Normalized compute budget for each subtask when generating $S = 45$ samples.

| Model | 2Wiki-R | 2Wiki-QA | Hot-R | Hot-QA | CWQ-R | CWQ-QA | WQSP-R | WQSP-QA | Decomp | ToolSel | ParamPred | Code | Static | Dynamic |
|---|---|---|---|---|---|---|---|---|---|---|---|---|---|---|
| Qwen2.5-72B | 2926 | 1174 | 2926 | 1174 | 1462 | 1174 | 1462 | 470 | 1462 | 4702 | 34942 | 17662 | 9022 | 4702 |
| Qwen2.5-32B | 1299 | 521 | 1299 | 521 | 649 | 521 | 649 | 209 | 649 | 2089 | 15529 | 7849 | 4009 | 2089 |
| Qwen2.5-7B | 283 | 112 | 283 | 112 | 140 | 112 | 140 | 45 | 140 | 455 | 3395 | 1715 | 875 | 455 |
| LLaMA-3.1-70B | 2845 | 1141 | 2845 | 1141 | 1421 | 1141 | 1421 | 457 | 1421 | 4571 | 33971 | 17171 | 8771 | 4571 |
| LLaMA-3.1-8B | 323 | 129 | 323 | 129 | 161 | 129 | 161 | 52 | 161 | 521 | 3881 | 1961 | 1001 | 521 |
| LLaMA-3.2-3B | 120 | 47 | 120 | 47 | 59 | 47 | 59 | 19 | 59 | 194 | 1454 | 734 | 374 | 194 |
| Gemma-2-27B | 1096 | 439 | 1096 | 439 | 547 | 439 | 547 | 176 | 547 | 1762 | 13102 | 6622 | 3382 | 1762 |
| Gemma-2-9B | 364 | 145 | 364 | 145 | 181 | 145 | 181 | 59 | 181 | 586 | 4366 | 2206 | 1126 | 586 |
| Gemma-2-2B | 79 | 31 | 79 | 31 | 39 | 31 | 39 | 13 | 39 | 129 | 969 | 489 | 249 | 129 |
| Phi-3-medium | 567 | 227 | 567 | 227 | 283 | 227 | 283 | 91 | 283 | 913 | 6793 | 3433 | 1753 | 913 |
| Phi-3-small | 283 | 112 | 283 | 112 | 140 | 112 | 140 | 45 | 140 | 455 | 3395 | 1715 | 875 | 455 |
| Phi-3-mini | 153 | 60 | 153 | 60 | 75 | 60 | 75 | 25 | 75 | 246 | 1842 | 930 | 474 | 246 |

Table 9: Normalized compute budget for each subtask when generating $S = 90$ samples.

| Model | 2Wiki-R | 2Wiki-QA | Hot-R | Hot-QA | CWQ-R | CWQ-QA | WQSP-R | WQSP-QA | Decomp | ToolSel | ParamPred | Code | Static | Dynamic |
|---|---|---|---|---|---|---|---|---|---|---|---|---|---|---|
| Qwen2.5-72B | 5086 | 2254 | 5086 | 2254 | 2542 | 2254 | 2542 | 902 | 2542 | 9022 | 69502 | 34942 | 17662 | 9022 |
| Qwen2.5-32B | 2259 | 1001 | 2259 | 1001 | 1129 | 1001 | 1129 | 400 | 1129 | 4009 | 30889 | 15529 | 7849 | 4009 |
| Qwen2.5-7B | 493 | 217 | 493 | 217 | 245 | 217 | 245 | 87 | 245 | 875 | 6755 | 3395 | 1715 | 875 |
| LLaMA-3.1-70B | 4945 | 2191 | 4945 | 2191 | 2471 | 2191 | 2471 | 877 | 2471 | 8771 | 67571 | 33971 | 17171 | 8771 |
| LLaMA-3.1-8B | 563 | 249 | 563 | 249 | 281 | 249 | 281 | 100 | 281 | 1001 | 7721 | 3881 | 1961 | 1001 |
| LLaMA-3.2-3B | 210 | 92 | 210 | 92 | 104 | 92 | 104 | 37 | 104 | 374 | 2894 | 1454 | 734 | 374 |
| Gemma-2-27B | 1906 | 844 | 1906 | 844 | 952 | 844 | 952 | 338 | 952 | 3382 | 26062 | 13102 | 6622 | 3382 |
| Gemma-2-9B | 634 | 280 | 634 | 280 | 316 | 280 | 316 | 112 | 316 | 1126 | 8686 | 4366 | 2206 | 1126 |
| Gemma-2-2B | 139 | 61 | 139 | 61 | 69 | 61 | 69 | 25 | 69 | 249 | 1929 | 969 | 489 | 249 |
| Phi-3-medium | 987 | 437 | 987 | 437 | 493 | 437 | 493 | 175 | 493 | 1753 | 13513 | 6793 | 3433 | 1753 |
| Phi-3-small | 493 | 217 | 493 | 217 | 245 | 217 | 245 | 87 | 245 | 875 | 6755 | 3395 | 1715 | 875 |
| Phi-3-mini | 267 | 117 | 267 | 117 | 132 | 117 | 132 | 47 | 132 | 474 | 3666 | 1842 | 930 | 474 |

## A.3 Dollar Costs of Repeated Sampling

Table 10 is the API cost information for each model from Together AI. We do not convert it in the same manner as above, as the dollar serves as a natural unit of price.

| Model Name | Parameters | Inference Cost (per 1M tokens) |
|---|---|---|
| Qwen2.5 72B | 72B | $1.20 |
| Qwen2.5 32B | 32B | $0.80 |
| Qwen2.5 7B | 7B | $0.30 |
| LLaMA-3.1-70B | 70B | $0.88 |
| LLaMA-3.1 8B | 8B | $0.18 |
| LLaMA-3.2 3B | 3B | $0.06 |

Table 10: Inference costs per 1M tokens for selected models from Together AI.

## A.4 Benchmarks, Datasets, Models, Metrics, and other Experimental Setup

We introduce the benchmarks, datasets, models, evaluation metrics, and other experimental setups.

**Retrieval-based Question Answering** uses two datasets: 2WikiMultiHopQA [11] and Hot-potQA [50], each providing 100 candidate text chunks per query. The task involves two subtasks: (1) retrieving relevant documents using an LLM retriever, and (2) answering the question using another LLM with the retrieved documents as context. For both datasets, the average prompt and generation lengths are 2048 and 128 for retrieval tasks, and 256 and 64 for the QA subtasks, respectively. We employ Qwen 2.5 (7B, 32B, 72B) [47] for retrieval and LLaMA-3 (3B, 8B, 72B) [7] for answering. These model families are selected due to their strong subtask performance and wide range of model sizes, which helps mitigate the influence of pre-training differences. This setup is used consistently across the other three tasks. The evaluation metrics are retrieval F1 (Ret-F1) and exact match (EM), corresponding to the two subtasks, respectively. For the Ret-F1 metric, given the ground-truth

retrieval documents, precision is computed as the fraction of correctly retrieved documents among all predicted documents, while recall is the fraction of ground-truth documents that are successfully retrieved. The F1 score is then calculated as the harmonic mean of precision and recall. For the EM metric, an answer is considered correct only if it exactly matches the ground truth. The EM score is computed as the fraction of such exactly matched answers among all predictions.

**Knowledge Graph Question Answering**    This task uses two KGQA datasets: ComplexWebQuestions [36] and WebQSP [51]. The objective is to retrieve relevant knowledge triplets from 100 candidates in the knowledge graph and answer the question based on the retrieved context. The task includes two subtasks: knowledge retrieval and question answering. For both datasets, the average prompt and generation lengths are 2048 and 64 for retrieval tasks, and 256 and 64 for the QA subtasks, respectively. We use Qwen 2.5 (7B, 72B) [47] for retrieval and LLaMA-3 (8B, 70B) [7] for answering. The evaluation metrics are retrieval F1 (Ret-F1) and exact match (EM), corresponding to the two subtasks, respectively. The computation of Ret-F1 and EM is the same as above.

**Task Automation**    This task involves decomposing complex user instructions into subtasks and invoking external tools to execute them. We use the TaskBench benchmark [32] to evaluate LLM performance across three stages: task decomposition, tool selection, and parameter prediction. The average prompt and generation lengths are 2048 and 64 for task decomposition, 1024 and 256 for tool selection, and 1024 and 2048 for parameter prediction. LLaMA-3 (8B, 70B) [7] is used for all subtasks. To evaluate LLMs' ability to understand and decompose complex tasks, we assess the quality of task decomposition using three complementary ROUGE metrics, including ROUGE-L. To evaluate tool selection, we introduce Node F1 (n-F1), which measures the correctness of predicted tools by comparing them against a reference set, reflecting the model's ability to identify appropriate tools for each subtask. To assess the accuracy of tool configuration, we use Parameter Name & Value F1 (p-F1), which evaluates both the identification of parameter names and the correctness of their assigned values, capturing the model's ability to provide contextually appropriate configurations.

**Automated Software Development**    We leverage ChatDev [29], a chat-driven framework based on the waterfall model, encompassing documentation, design, coding, static testing, and dynamic testing phases. Our focus is on three core subtasks: coding, static testing, and dynamic testing. The average prompt and generation lengths are 1024 and 1024 for coding, 1024 and 512 for static testing, and 1024 and 256 for dynamic testing. We use LLaMA-3 (3B, 70B) [7] for all subtasks. Consistency measures how well the generated software code aligns with the original requirement description. It is quantified as the cosine similarity between the semantic embeddings of the textual requirements and the generated code. A higher score indicates a greater degree of consistency with the specified requirements.

**Other Experimental Setup**    To encourage diverse generations and activate LLMs' coverage during repeated sampling, we set the temperature to $0.9$, while keeping all other decoding hyperparameters at their default values. Each trial is conducted using an NVIDIA H100 80GB HBM3 GPU to ensure consistent runtime performance across all datasets. For each dataset, we randomly sample 50 instances to form the training set used for search and 500 instances as the test set for final evaluation. All datasets used in our experiments are made publicly available through an anonymous link to ensure transparency and reproducibility.

### A.5    Model Selection for Subtask-Specific Analysis

For each subtask, we select models from the same model family and assume that model size reflects model capacity. By comparing the performance of small and large models within the same family under test-time scaling, we can figure out whether smaller models can outperform larger ones under equivalent compute budgets. This comparison focuses solely on the relationship between test-time compute and model size. Otherwise, comparisons across different model families could introduce confounding effects due to variations in pretraining, such as differences in pre-training data or objectives. Therefore, we restrict each subtask to models within a single family. During evaluation, all other non-target subtasks are fixed to use LLaMA-3 70B with one-pass inference to isolate the effect of model choice in the target subtask. For each subtask, we select the model family that achieves the best performance. Fig. 8a-8i present subtask-level performance across model sizes and families. We exclude ChatDev from this analysis, as it does not provide metrics for intermediate steps.

Based on the results, we select the Qwen family for retrieval-related subtasks and the LLaMA-3 family for all others.

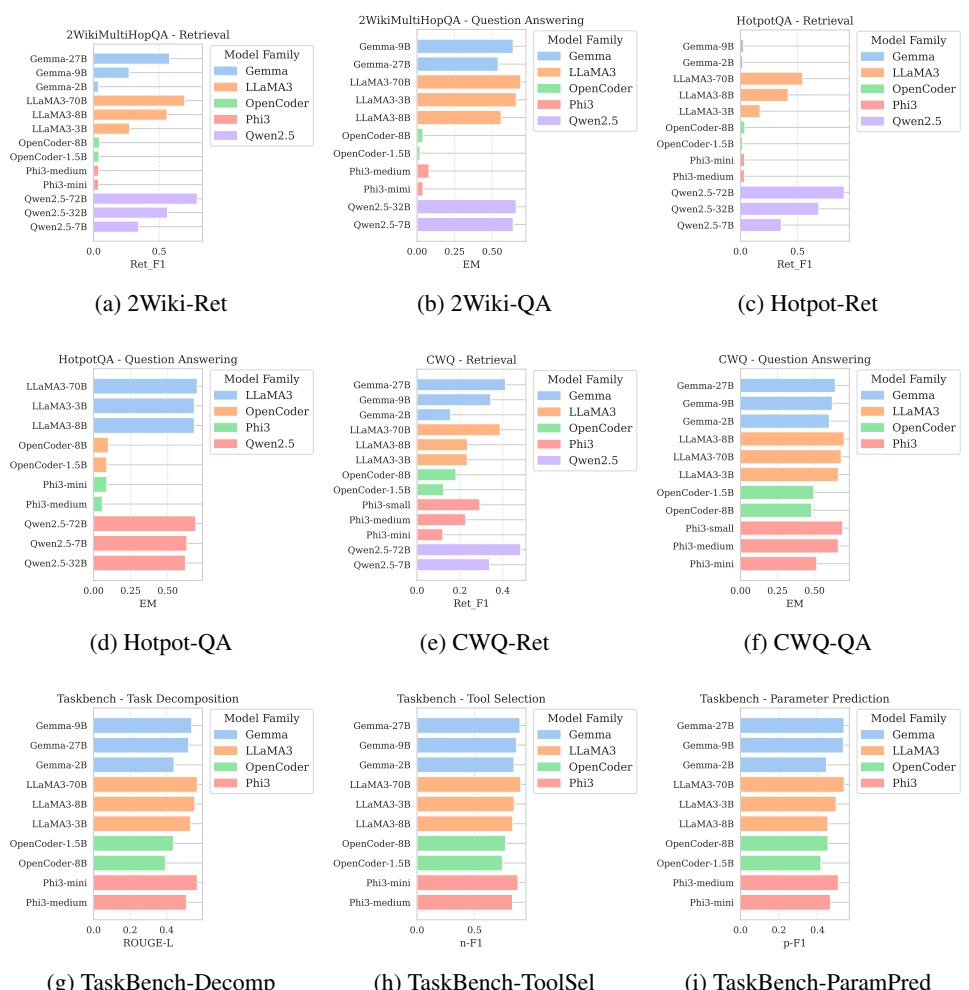

Figure 8: Performance of various LLMs across retrieval, question answering, and task execution subtasks on three types of tasks.

## A.6 More Preliminary Experimental Results

Fig. 11, 12, 13, and 14 illustrate how subtask performance evolves in KGQA, TaskBench, and ChatDev as the test-time sampling, FLOPs, and budget increase.

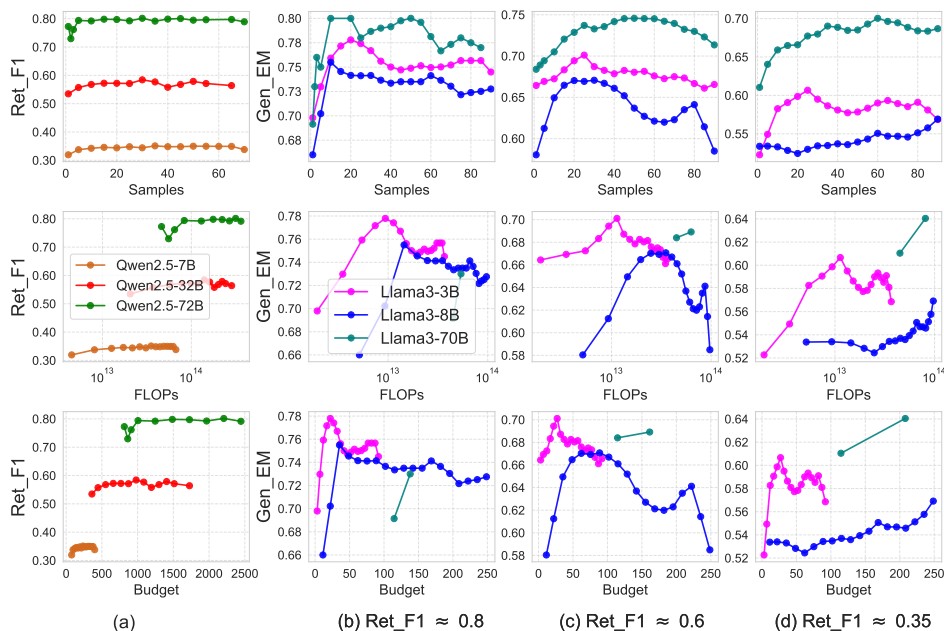

Figure 9: Performance variance on 2WikiMultiHopQA with increasing sampling, inference FLOPs, and budget. Top: performance by sample count; middle: performance by log-scaled inference FLOPs; bottom: performance by normalized budget. (a) Retrieval accuracy measured by Retrieval F1 (Ret_F1). (b-d) QA performance under varying retrieval quality levels measured by Gen_EM, the exact match between the generated answer and the ground truth.

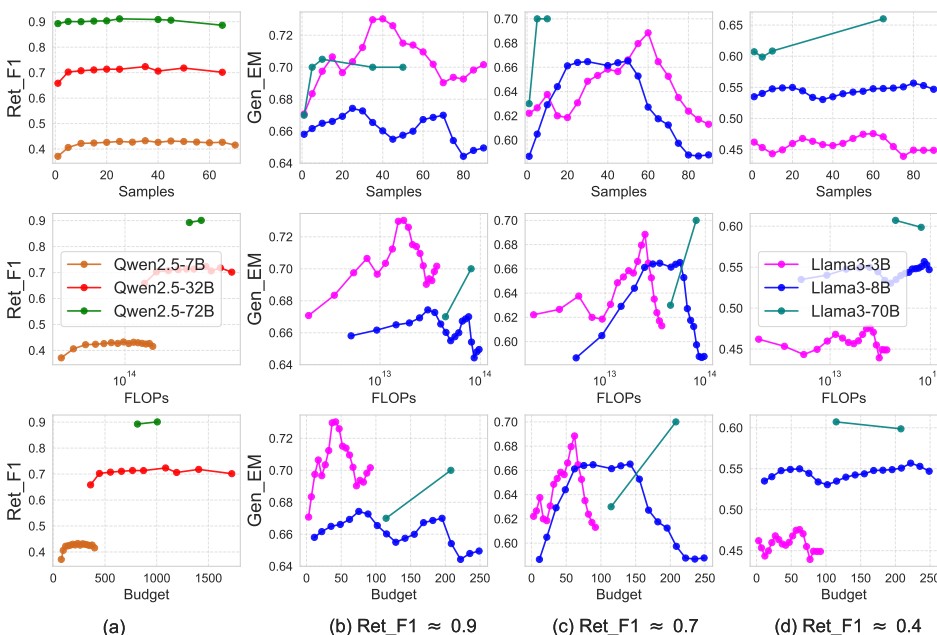

Figure 10: Performance variance on HotpotQA with increasing sampling, inference FLOPs, and budget. Top: performance by sample count; middle: performance by log-scaled inference FLOPs; bottom: performance by normalized budget. (a) Retrieval subtask performance; (b-d) Question answering performance under different retrieval quality levels.

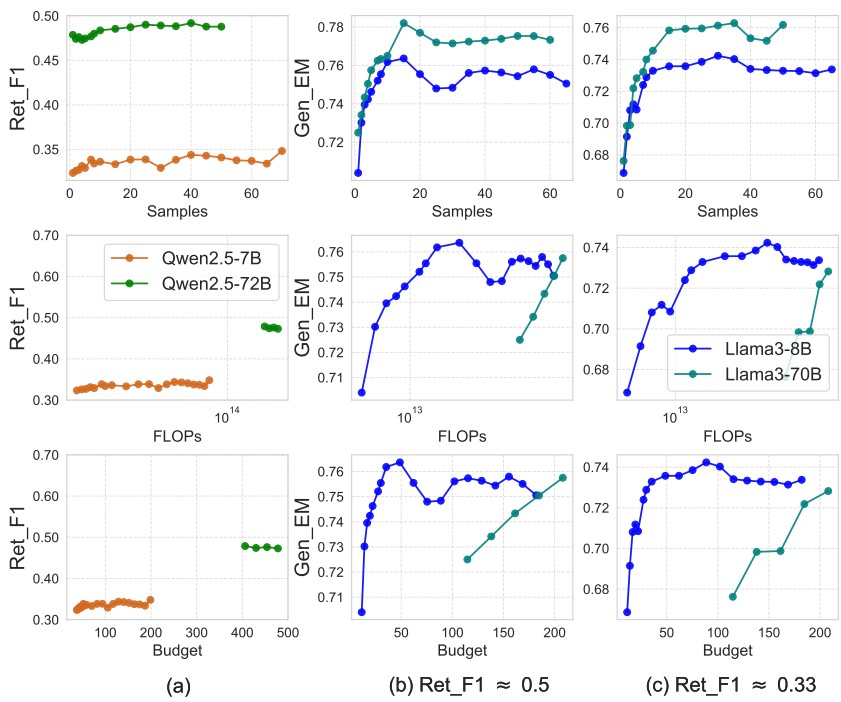

Figure 11: Performance variance on CWQ with increasing sampling, inference FLOPs, and budget. Top: performance by sample count; middle: performance by log-scaled inference FLOPs; bottom: performance by normalized budget. (a) Retrieval performance; (b-d) Question answering performance under different retrieval quality levels.

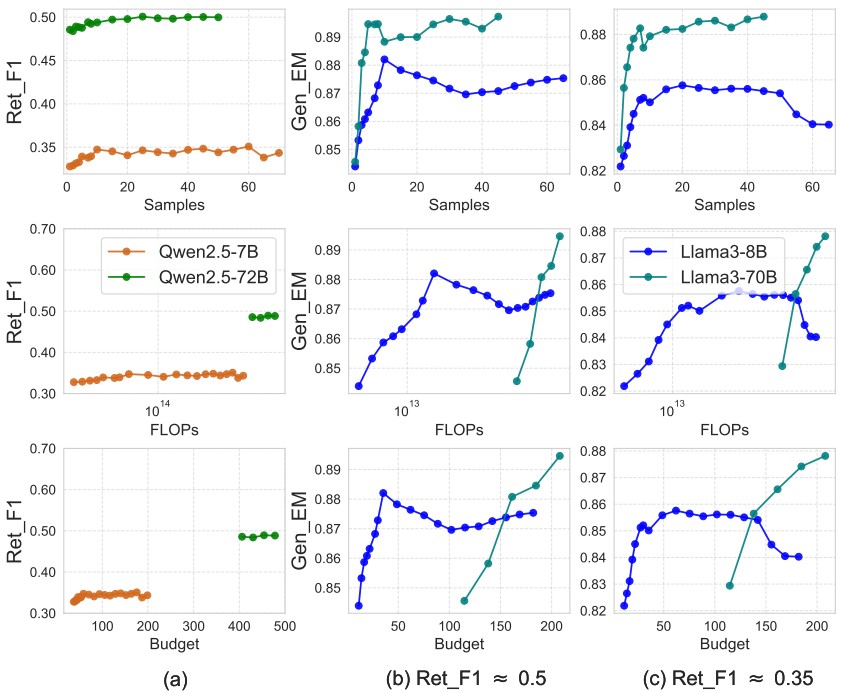

Figure 12: Performance variance on WebQSP with increasing sampling, inference FLOPs, and budget. Top: performance by sample count; middle: performance by log-scaled inference FLOPs; bottom: performance by normalized budget. (a) Retrieval performance; (b-d) Question answering performance under different retrieval quality levels.

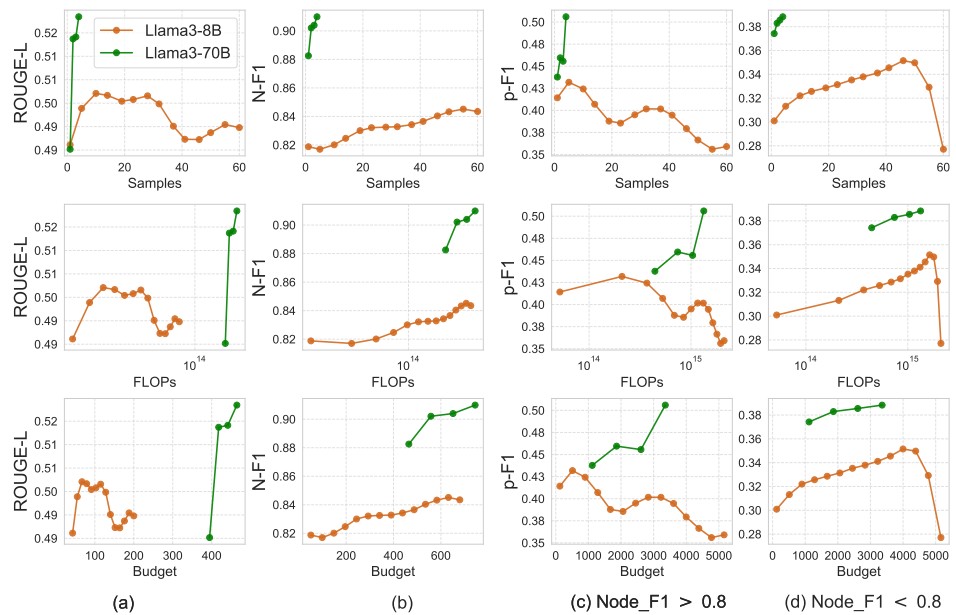

Figure 13: Performance variance on Taskbench-DailyAPIUsage with increasing sampling, inference FLOPs, and budget. Top: performance by sample count; middle: performance by log-scaled inference FLOPs; bottom: performance by normalized budget. (a) Task decomposition performance; (b) Tool selection performance; (c-d) Parameter prediction under different tool-selection quality levels.

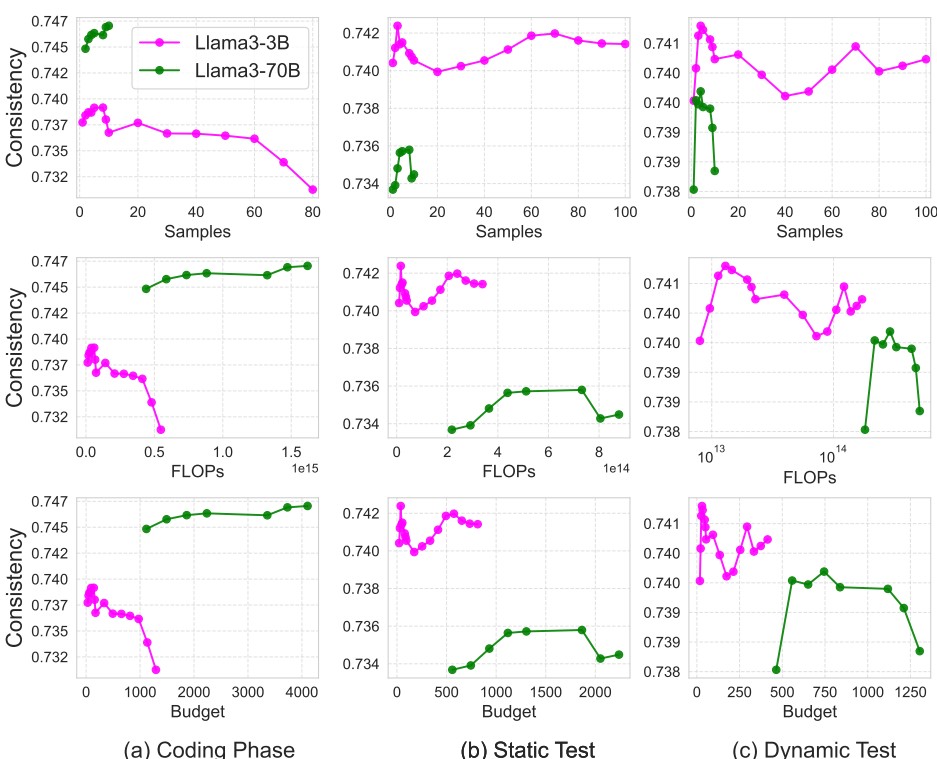

Figure 14: Performance variance on ChatDev with increasing sampling and inference FLOPs. Top: performance by sample count; middle: performance by log-scaled inference FLOPs; bottom: performance by normalized budget. (a-c) The consistency score during coding, static test, and dynamic test, respectively.

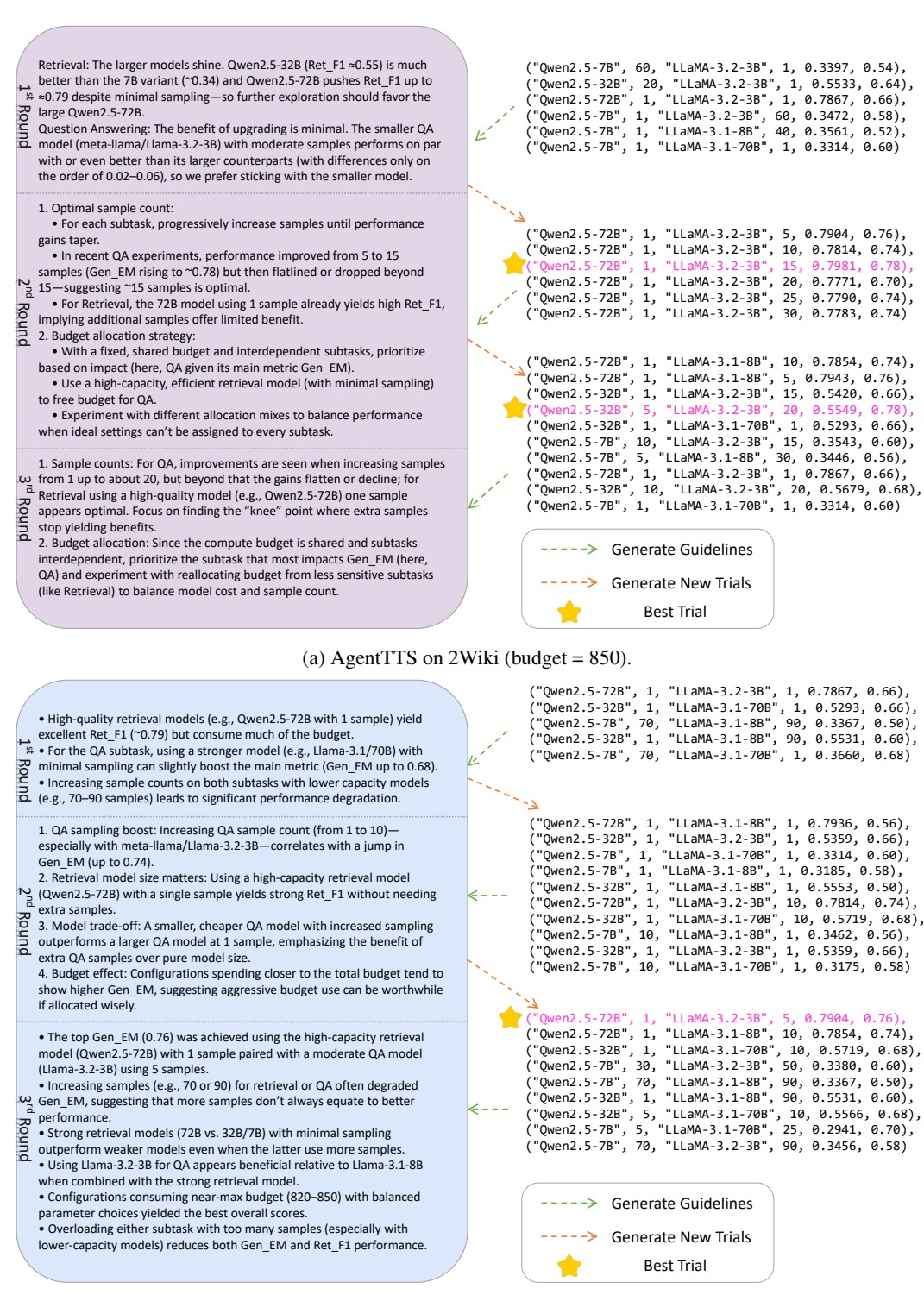

(a) AgentTTS on 2Wiki (budget = 850).

(b) AgentHPO on 2Wiki (budget = 850).

Figure 15: Comparison of complete trial generation and decision guidelines using AgentTTS and AgentHPO on the 2Wiki dataset under a compute budget of 850.

## A.7    Prompt Design in AgentTTS for Search Guidelines and Trial Generation

We design prompts below to guide the LLM agent in generating both guidelines and candidate trials. Each prompt is constructed to provide the agent with basic task information and requirements, enabling effective reasoning and planning. For guideline generation, we distinguish between the initial and non-initial phases. The initial guideline prompt, informed by Insight 1, helps identify whether each subtask favors smaller or larger models, thereby guiding early-stage search. The non-initial guideline prompt, informed by Insights 2 and 3, reflects general test-time scaling patterns and assists in balancing budget allocations across subtasks. For trial generation, the LLM agent proposes new configurations based on the existing guidelines.

---

**Fusion Prompt**

You have been provided with a set of responses from various open-source models to the latest user query, which is query. Your task is to synthesize these responses into a single, high-quality response. It is crucial to critically evaluate the information provided in these responses, recognizing that some of it may be biased or incorrect. Your response should not simply replicate the given answers but should offer a refined, accurate, and comprehensive reply to the instruction. Ensure your response is well-structured, coherent, and adheres to the highest standards of accuracy and reliability. Once again, the query is: query
Responses from models:

---

**LLM Prompt for Initial Guidelines**

[System Prompt] You are an expert in parameter optimization.
[User Prompt] I am tuning test-time parameters for a complex task that can be broken down into multiple simple subtasks. The parameters include the model size and the number of samples used for each subtask. Generally, increasing the number of samples improves performance. The goal is to find the best parameter settings that maximize performance within a fixed budget.
I will provide the task name, description, budget, and initial search history. Please help me summarize the guidelines from the search history in the following aspect:
1. The search history includes initial trials that use different model sizes under the same unit budget. In these trials, smaller models are allowed more samples, while larger models have fewer samples due to their higher cost. Please help me identify which model size we should explore further in each subtask. If the large model performs significantly better than the small model, we will choose the large one. Otherwise, if its performance is similar or only slightly better, we will prefer the smaller model.
2. As the number of samples increases, performance tends to improve and then level off. Please identify the search direction for finding the optimal number of samples for each subtask, especially the point after which more samples no longer improve results.
3. The total compute budget is shared across all subtasks and they are interdependent. If we could not assign the optimal model and sample count to every subtask, consider the following strategy. Leveraging your planning capability, prioritize each subtask, try different combinations of budget allocations for the other tasks, and balance the allocations across subtasks.
Inputs:
• Task name: {task_name}
• Task description: {task_desc}
• Subtasks, models, prompt/generation lengths: {subtask_specification}, {model_space}
• Total budget: {budget}
• Evaluation metrics: {metrics}
• Main metric: {main_metric}
• Search history: {history}
Please generate your response in the most concise wording.
Response:

---

## LLM Prompt for Non-Initial Guidelines

[System Prompt] You are an expert in parameter optimization.

[User Prompt] I am tuning test-time parameters for a complex task that can be broken down into multiple simple subtasks. The parameters include the model size and the number of samples used for each subtask. Generally, increasing the number of samples improves performance. The goal is to find the best parameter settings that maximize performance within a fixed budget.

I will provide the task name, description, budget, and both old and recent search history.

Please help me summarize guidelines from the recent search history in the following three aspects:

1. As the number of samples increases, performance tends to improve and then level off. Please identify the search direction for finding the optimal number of samples for each subtask, especially the point after which more samples no longer improve results.

2. The total compute budget is shared across all subtasks, and they are interdependent. If we could not assign the optimal model and sample count to every subtask, consider the following strategy. Leveraging your planning capability, prioritize each subtask, try different combinations of budget allocations for the other tasks, and balance the allocations across subtasks.

3. If you observe that the performance variance across the search history is small, please suggest that future searches focus more on exploration strategies such as crossover, mutation, and random search, rather than strictly following the existing search patterns.

Inputs:
- Task name: {task_name}
- Task description: {task_desc}
- Subtasks, models, prompt/generation lengths: {subtask_specification}, {model_space}
- Total budget: {budget}
- Evaluation metrics: {metrics}
- Main metric: {main_metric}
- Search history: {history}

Please generate your response in the most concise wording.

Response:

## LLM Prompt for Trial Generation

[System Prompt] You are an expert in parameter optimization.

[User Prompt] I am tuning test-time parameters for a complex task decomposable into multiple subtasks. Each subtask requires selecting a model and the number of samples. Increasing the number of samples generally improves performance. The objective is to identify configurations that maximize overall performance under a fixed compute budget.

I will provide the task name, task description, a list of subtasks, and the model choices available for each, the total budget, available samples, the budget calculation formula, evaluation metrics (including the main metric to optimize), search history (previous configurations tried), and insights from previous experiments.

The budget for each subtask is computed with the following Python function:

```python
def compute_budget(num_samples, model_size, prompt_length, generation_length,
                   M_small=3e9, Np2=128, Nd2=64):
    alpha = model_size / M_small
    beta1 = prompt_length / generation_length
    beta2 = prompt_length / Np2
    beta3 = Np2 / Nd2

    budget = beta3 * ((alpha * beta2 / beta1) * (beta1 + num_samples) - 1)
    return budget
```

The total budget is the sum across all subtasks. Using the information provided, generate {batch_size} candidate configurations that follow the insights and stay within the budget. Return exactly {batch_size} new configurations in strict JSON format, following the schema:

{ "subtask_1": { "model": "model_name", "samples": int }, ... }

Inputs:
- Task name: {task_name}
- Task description: {task_desc}
- Subtasks, models, prompt/generation lengths: {subtask_specification}, {model_space}
- Total budget: {budget}
- Evaluation metrics: {metrics}
- Main metric: {main_metric}
- Search history: {history}
- Guidelines: {experience}

Response: Return only {batch_size} candidates in strict JSON format.

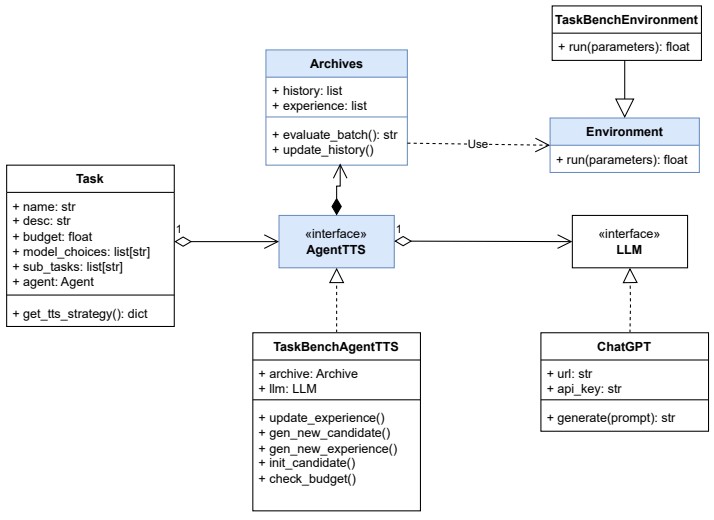

Figure 16: Class Diagram of the AgentTTS Framework.

## A.8 Detailed Cases for Interpretability

We present a complete case study comparing AgentTTS and AgentHPO in Fig. 15. The budget of 850 poses a challenge in balancing optimal usage across two subtasks, as the optimal retrieval configuration (Qwen2.5-72B, 1 sample) consumes 814 units, while the optimal QA setting (LLaMA-3-3B, 50 samples) requires 52. We observe: (1) AgentTTS identifies optimal configurations using embedded insights, whereas AgentHPO does not; (2) Insights 1, 2, and 3 guide different stages of the search: Insight 1 selects models (Qwen2.5-72B for retrieval, LLaMA-3-3B for QA), Insight 2 focuses sampling (reaching the optimal trial: Qwen2.5-72B, 1 / LLaMA-3-3B, 15), and Insight 3 balances budget use (yielding another optimal trial: Qwen2.5-32B, 5 / LLaMA-3-3B, 20).

## A.9 Baselines

We compare two baseline categories: LLM-based approaches (AgentHPO [20], MLCopilot [56], and LLM_ZS) and traditional methods (Random Search and Bayesian Optimization [31]).

**AgentHPO [20]**    targets Hyperparameter Optimization (HPO), addressing limitations of traditional AutoML methods such as high trial cost, complex setup, and limited interpretability by leveraging LLM-powered autonomous agents. It comprises two modules: the *Creator*, which converts natural language inputs (e.g., dataset, model, and goals) into initial hyperparameters, and the *Executor*, which trains models, logs outcomes, and returns feedback. The Creator iteratively refines HPs using this feedback, forming a closed-loop optimization. To adapt AgentHPO for our test-time scaling budget allocation setting, we provide AgentHPO with structured inputs similar to our method, including task descriptions, model choices, and total budget. However, AgentHPO assumes LLMs possess sufficient ML knowledge for HPO tasks, an assumption that fails in multi-stage test-time scaling, where domain-specific knowledge is lacking. Our method addresses this gap by injecting targeted insights about test-time scaling into the agent's reasoning process, improving both search efficiency and decision interpretability.

**MLCopilot [56]**    focuses on generating machine learning solutions, addressing limitations in traditional AutoML methods, such as Bayesian optimization [31], which often lack interpretability and struggle to incorporate human prior knowledge, e.g., regarding model architectures. Inspired by human design patterns, where individuals understand a task by recalling relevant experiences or knowledge from prior work or textbooks, MLCopilot emulates this process by leveraging LLMs to suggest solutions for new ML tasks based on historical cases. To adapt MLCopilot to our problem setting, we provide the search history of a similar task to guide the LLM in generating both prior experience and candidate trials iteratively for the target multi-stage task. However, while MLCopilot can extract surface-level patterns from related tasks, it lacks the ability to derive tailored insights necessary for efficient budget allocation in test-time scaling. This limits its effectiveness in the test-time scaling compute-optimal budget allocation problem, where fine-grained domain understanding is essential for optimal resource allocation.

**LLM_ZS**  LLM_ZS directly prompts the LLM to generate zero-shot candidate trials across multiple iterations, using only the task description, model choices, and total budget as input. Unlike methods that leverage prior knowledge or historical search experience, LLM_ZS relies solely on the LLM's pretrained knowledge to guide trial generation.

**Bayesian Optimization (BO) [31]**  is a model-based method for optimizing black-box functions and is widely used in hyperparameter tuning. For test-time scaling in multi-stage tasks, BO models the performance of a candidate allocation as a black-box objective and employs a surrogate model (e.g., Gaussian Process) to guide the search via acquisition functions (e.g., Expected Improvement). It balances exploration and exploitation to navigate the combinatorial space of model-budget configurations. However, its limited interpretability hinders its applicability. Moreover, in multi-stage tasks with complex subtask interactions and non-smooth performance landscapes, BO may converge to local optima and require a large number of evaluations.

**Random Search**  serves as a simple yet widely used baseline for hyperparameter tuning. In the context of test-time scaling budget allocation for multi-stage settings, it randomly samples model-budget configurations without relying on performance history or prior knowledge. Despite its simplicity, Random Search is often robust to non-smooth landscapes, which allows it to occasionally find globally optimal trials. However, its inherent randomness and lack of guidance result in significantly slower convergence compared to more informed or tailored strategies.

### A.10   Detailed Related Work

### A.10.1   Test-time Scaling and Compute-optimal Strategy

Inspired by the human reasoning process that tends to engage in deeper, more deliberate thinking, several studies have proposed allocating additional compute during inference to enhance task performance [44, 42]. Further, other works [2, 45] have observed that increasing inference-time compute exhibits a similar pattern akin to training scaling laws: additional inference compute consistently leads to improved task performance. This phenomenon is commonly referred to as *test-time scaling (TTS)*. Test-time scaling technologies can be mainly classified into two categories: **sequential scaling** and **parallel scaling**. In **sequential scaling**, it enhances test-time computation by generating progressively longer solutions along the sequence dimension. The most prevalent method is self-revision, where first generate an initial response and then iteratively evaluate and refine it based on self-assessment [24, 6, 33]. Because sequential scaling relies on the model's ability to generate reasonably accurate initial responses, it is generally more effective for simpler tasks [33]. On the other hand, **parallel scaling** generates multiple independent solutions in parallel and aggregates them into a final answer. Common solution-level aggregation methods are Best-of-N [35, 9], which sample N complete solutions and then select the best one according to a verifier. While Tree-Search algorithms, viewed as parallel scaling at the token or step level, are against a process-based reward model to search top-K intermediate steps and further explore, including Beam Search [53, 46] and Monte Carlo Tree Search (MCTS) [45, 33, 10, 37, 5, 59]. However, they typically rely on explicit guidance signals for candidate selection. An alternative line of work directly employs an LLM as the fuser to aggregate candidate solutions, offering greater generalization and flexibility [13, 17, 30]. Zeng et al. [55] argues that parallel scaling provides better coverage and scalability than sequential scaling, as extended sequential reasoning chains may corrupt intermediate correct outputs. Similarly, Snell et al. [33] demonstrates that parallel scaling is more suitable for complex tasks, as it requires only the ability to generate correct final answers. Therefore, we adopt a parallel test-time scaling strategy—specifically, **repeated sampling with fusion**, to address complex multi-stage reasoning tasks, leveraging the strengths of small models in diverse sampling and robust aggregation.

The configuration of test-time scaling strategies plays a critical role in performance improvement, giving rise to the broader problem of optimally allocating compute and budget, commonly referred to as the *test-time compute-optimal scaling strategy*. Recent studies [2, 45, 19, 54, 33, 41, 40] have shown that, in certain settings, smaller models can outperform large language models under the same compute budget. These works explore how to choose between training scaling and test-time scaling, as well as how to select among different test-time scaling techniques to optimize performance. For instance, Snell et al. [33] demonstrate that task difficulty strongly influences the optimal scaling strategy: for moderately difficult tasks, parallel search with small models is preferred, while for simpler tasks, sequential scaling with large models is more effective. Based on this observation,

they propose a difficulty predictor to guide scaling decisions dynamically. Liu et al. [19] further extend this line of work by showing that the choice of optimal TTS strategies is highly sensitive to the design of reward functions. Yue et al. [54] propose using a linear model to fit key factors influencing strategy selection within retrieval-augmented generation (RAG) systems. Wu et al. [45] introduce Reward Balanced Search (REBASE), a novel tree search algorithm that combines weighted voting to achieve a Pareto-optimal trade-off between accuracy and inference cost. However, these approaches primarily focus on single-stage task scenarios and do not systematically address the challenge of budget allocation across subtasks in multi-stage complex tasks. In this work, we extend the test-time compute-optimal scaling framework to multi-stage complex tasks.

### A.10.2   LLM for Hyperparameter Optimization

LLMs have emerged as promising tools for enhancing the efficiency and effectiveness of hyperparameter optimization (HPO) by leveraging contextual understanding and prior knowledge [8, 43, 61, 22, 38], compared to traditional Automated Machine Learning (AutoML) approaches such as Bayesian Optimization (BO) [31]. Existing LLM-based HPO studies can be mainly categorized into two directions: (1) using LLMs to reduce the search space for traditional methods, and (2) enabling LLMs to autonomously search optimal hyperparameters. On one hand, LLMs can significantly reduce the vast search spaces in Neural Architecture Search (NAS) and HPO [52, 25, 23, 20]. For example, GPT-NAS [52] combines GPT models with evolutionary algorithms [60] to rapidly prune low-quality architectures, enhancing search efficiency. Morris et al. [25] introduce "Evolution of Thought," allowing LLMs to refine architectures and hyperparameters based on feedback iteratively. AutoM3L [23] integrates external tools such as Ray.Tune for hyperparameter tuning, with LLMs recommending optimized search spaces. Llambo [20] formulates BO problems in natural language, enabling LLMs to propose high-potential candidates and iteratively optimize based on historical observations.

On the other hand, several works [4, 18, 62, 58, 1, 12, 56] empower LLMs to search hyperparameter configurations autonomously. AutoMMLab [49] and GENIUS [62] treat LLMs as black-box optimizers, iteratively refining configurations based on evaluation feedback. CAAFE [12] focuses on tabular data, where LLMs generate new semantically meaningful features based on dataset descriptions. Moreover, LLM-based agent frameworks have emerged, leveraging feedback from machine learning platforms or historical experiments. For instance, AutoML-GPT [58] automates the pipeline from preprocessing to model selection and hyperparameter tuning, adjusting strategies based on experimental logs. MLCopilot [56] predicts hyperparameters for unseen tasks by canonicalizing prior task experiences and interactively refining solutions through LLM reasoning. AgentHPO [20] autonomously processes task descriptions, conducts experiments, and iteratively improves hyperparameter search based on accumulated trials. Building upon these advancements, we extend the LLM-agent framework to the domain of *test-time scaling budget allocation in multi-stage complex tasks*, aiming to search for compute-optimal scaling configurations.

### A.11   Limitations

Our problem focuses on multi-stage tasks with static stages, where the compute budget is allocated across a fixed set of subtasks. However, in real-world applications, some tasks exhibit dynamic multi-stage behavior, where the actual runtime subtasks may vary depending on input conditions or user interaction. For example, in voice-based personal assistants on mobile devices, the system may dynamically decide whether to perform document retrieval, clarification, or follow-up question generation based on the user's query and context. In such scenarios, it is difficult to predefine all possible subtasks and their corresponding budget allocations. Consequently, applying our LLM-agent-based search framework to these dynamic multi-stage tasks presents a considerable challenge.

### A.12   Broader Impact

This work proposes AgentTTS, an LLM-agent-based framework for compute-optimal test-time scaling in multi-stage tasks. The methodology offers significant positive societal impacts. First, by improving inference efficiency and reducing unnecessary computation, it contributes to more sustainable and cost-effective deployment of large language models, especially in resource-constrained environments. Second, our framework promotes better alignment between model selection and

task complexity, enabling more accessible AI solutions across diverse application domains such as education, healthcare, and low-resource software development.

Nonetheless, our approach raises potential concerns. AgentTTS relies on repeated sampling from foundation LLMs, which can amplify not only their strengths but also their limitations. For instance, hallucinations in large language models may be intensified through test-time scaling integration and efficiently propagated, increasing the risk of misinformation. Moreover, LLMs are susceptible to adversarial attacks such as jailbreaks, backdoor injections, and membership inference attacks, thereby heightening the security risks associated with AgentTTS.

