# OpenReview forum: "AgentTTS: Large Language Model Agent for Test-time Compute-optimal Scaling Strategy in Complex Tasks"
_NeurIPS.cc/2025/Conference — NeurIPS 2025 poster_

### Official Review · Reviewer_NhkM · 2025-06-30

**Clarity:** 2
**Significance:** 3
**Originality:** 3
**Rating:** 4
**Confidence:** 3

**Summary:**

This paper proposes AgentTTS,  an LLM-agent framework to efficiently allocate compute resources across subtasks in multi-stage complex tasks. This framework aims to solve test-time compute-optimal scaling in multi-stage complex tasks. By conducting empirical studies on four task types and six datasets, the authors identify three key insights—subtasks have distinct model preferences, test-time scaling yields diminishing returns, and early subtask budgets affect downstream performance—which are then integrated into AgentTTS’s iterative, feedback-driven search process. Experiments show that AgentTTS outperforms both traditional and LLM-based baselines in search efficiency, robustness, and interpretability for compute-optimal scaling in complex workflows.

**Questions:**

See weakness

**Ethical Concerns:**

["NO or VERY MINOR ethics concerns only"]

**Final Justification:**

Authors's response address most of my concerns. I have updated the score accordingly.

**Limitations:**

yes

**Quality:**

3

**Strengths And Weaknesses:**

Strength:
1. The paper addresses a novel and highly relevant problem, i.e. the test-time compute-optimal scaling in multi-stage complex tasks, aiming to select suitable models and allocate budgets per subtask to maximize overall performance.
2. The design of AgentTTS is well-motivated by three empirical insights derived from preliminary experiments.
3. Authors conduct experiments across four task types and six datasets, demonstrating the superiority of the proposed framework.

Weakness:
1. While the empirical insights are valuable, the paper lacks deeper theoretical understanding of why these patterns emerge. The insights appear somewhat ad-hoc without principled theoretical foundations explaining the underlying mechanisms driving these behaviors.
2. The three insights are derived from specific task types and may not generalize to other multi-stage scenarios. The paper doesn't adequately address the boundaries of applicability for these insights.
3. The FLOPs-based budget conversion assumes linear scaling relationships that may not hold across all model architectures and task types. Also, the choice of base configuration (3B model, 128 prompt tokens, 64 generation tokens) seems arbitrary and may bias results.
4. It seems only GPT-o3-mini is used as the agent LLM, the paper could benefit from applying the method on more LLMs to show its generalizability.

---

> ### Author Rebuttal · Authors · 2025-07-31
>
> We thank the reviewer for valuable feedback on theory and generalization. Below we address each point.
>
> > W1 ... the paper lacks deeper theoretical understanding of why these patterns emerge. The insights appear somewhat ad-hoc without principled theoretical foundations explaining the underlying mechanisms driving these behaviors.
>
> Thank you to the reviewer for highlighting the lack of in-depth theoretical analysis of our proposed insights. In response, we provide a complete theoretical analysis below.
>
> **Setup.** Model $\pi$ receives input $x$, samples a reasoning prefix $r \sim \pi(\cdot \mid x)$, then an answer $y \sim \pi(\cdot \mid x + r)$.  Let the correct answer be $Y^\star$. The single-sample success is
> $$
> p_\pi(x) = \Pr_\pi(y = Y^\star \mid x) = E_{r \sim \pi(\cdot \mid x)}\big[\pi(Y^\star \mid x + r)\big].
> $$
> With $k$ samples, we obtain $(r_{1:k}, y_{1:k})$ and an aggregator $A_k$ outputs the final prediction $y^ \star = A_k(y_{1:k}, r_{1:k}, x)$ (abbreviated as $\pi(y^\star \mid y_{1:k})$), so we have
> $$
> p_\pi^k =\Pr_\pi(y^ * =Y^*|x) = E_{r\sim \pi(\cdot|x), y_{1:k}}[\Pi_{i\in[1,k]} \pi(y_i|x+r) \pi(Y^ * |y_{1:k})]
> $$
>
> **Insight 1: subtask-specific preference on model and on $k$**
>
> **Model preference:** Define the per-sample log-odds (against the strongest competitor $\bar{Y}$):
> $$
> L(r; x, \pi) = \log \frac{\pi(Y^\star \mid x + r)}{\pi(\bar{Y} \mid x + r)}, \qquad
> \gamma_\pi(x) = E_r[L(r; x, \pi)].
> $$
> $\gamma_\pi(x)$ depends on both aspects:  (i) $\pi(r \mid x)$ producing helpful $r$, and (ii) $\pi(y \mid x + r)$ separating $Y^\star$ from others. Different subtasks $i$ require different capabilities, so $\gamma_{\pi,i}(x)$ varies across subtasks, leading to different preferred models.
>
> **The preferred number of samples is subtask- and model-specific:** We define $k^ *$ as the optimal number of samples for a given subtask. Its value depends on two key factors: (i) the correctness of individual outputs $y_i$, and (ii) the accuracy of the aggregation function $\pi(Y^ * \mid y_{1:k})$. Harder subtasks generally require larger $k$ due to lower per-sample accuracy, while the effectiveness of aggregation is constrained by the model’s long-context handling ability. As a result, the optimal sampling count $k^ *$ varies across subtasks and models.
>
> **Insight 2: there is an optimal $k$**
>
> Repeated sampling helps but saturates with correlation and can worsen due to long-context aggregation.  According to Hoefffding inequality, we could get a simple composite view:
> $$\Pr(y^ \star \ne Y^ \star) \lesssim \exp(-\alpha k) + C(k), \alpha = 2(2p_\pi(x) - 1)^2.
> $$
>
> $C(k)$ is small for $k \le k_0$ and then increases (e.g., due to distraction or truncation).  The sum is U-shaped, so
>
> $$
> k^\star = \arg\min_k { \exp(-\alpha k) + C(k) }
> $$
>
> explains the “improve $\to$ plateau $\to$ degrade” pattern as $k$ increases.
>
> **Insight 3: earlier subtasks affect later model and $k$**
>
> Let $z$ be upstream output appended to the context; the downstream stage uses $(x, z)$. Then:
> $$
> \gamma_\pi(x, z) \ge \gamma_\pi(x), \qquad p_\pi(x, z) \ge p_\pi(x),
> $$
> so the downstream optimal number of repeats drops: $k^\star(x, z) < k^\star(x)$.
>
> Moreover, informative $z$ can make a smaller downstream model adequate:
> $$
> \gamma_{\pi_{\text{small}}}(x, z) \approx \gamma_{\pi_{\text{large}}}(x),
> $$
> with much lower cost—changing the preferred downstream model.
>
> Let $F$ denote the expected task score. If
> $$
> \frac{\partial^2 F}{\partial B_{\text{up}} \partial B_{\text{down}}} < 0,
> $$
> then upstream and downstream budgets are substitutes:  more upstream budget reduces the downstream best-response budget and can flip the downstream argmax over models.
>
>
> > W2 The three insights are derived from specific task types and may not generalize to other multi-stage scenarios. The paper doesn't adequately address the boundaries of applicability for these insights.
>
> We thank the reviewer for raising the important point.
>
> To assess generalizability, we evaluated our framework and validated the proposed insights across **four diverse task types** spanning **six benchmark datasets**, introduced in Appendix A.4, including:
>
> - **Open-domain question answering** (e.g., 2WikiMultiHopQA, HotpotQA),
> - **Knowledge graph question answering** (e.g., CWQ, WebQSP),
> - **Task Automation** (e.g., Taskbench),
> - **Automated Software Development** (e.g., ChatDev).
>
> These tasks exhibit different multi-stage structures and model behaviors.  In the **Appendix A.6**, the consistency of the observed trends across these varied domains provides strong empirical support that the proposed insights are **not specific to a single task type** but rather **reflect common patterns** in multi-stage complex tasks.
>
> To further address the concern about generalization to other tasks, we also adapted the **TextVQA** dataset into a two-stage pipeline: **Stage 1**: Image understanding; **Stage 2**: Question-specific reasoning based on image output.
>
> We then evaluated the **test-time scaling behavior** and **search efficiency** for multiple models in this multimodal setting. Results are shown below.
>
> **Table: Llama-3.2-11B-Vision-Instruct Summarizes Images**
>
> |Sample|1|5|10|15|20|25|30|35|40|45|50|55|60|65|70|80|90|95|
> |------|--|--|---|---|---|---|---|---|---|---|---|---|---|---|---|---|---|---|
> |Budget|20.0|49.33|86.0|122.67|159.33|196.0|232.67|269.33|306.0|342.67|379.33|416.0|452.67|489.33|526.0|599.33|672.67|709.33|
> |Accuracy|0.6954|0.7605|0.8419|0.8368|0.8396|0.8423|0.8421|0.8351|0.8264|0.8240|0.8289|0.8338|0.8310|0.8355|0.8400|0.8289|0.8254|0.8249|
>
> **Table: Llama-3.2-90B-Vision-Instruct Summarizes Images**
>
> |Sample|1|5|10|15|20|25|30|35|45|50|55|60|65|80|95|
> |------|-|-|--|--|--|--|--|--|--|--|--|--|--|--|--|
> |Budget|178.0|418.0|718.0|1018.0|1318.0|1618.0|1918.0|2218.0|2818.0|3118.0|3418.0|3718.0|4018.0|4918.0|5818.0|
> |Accuracy|0.6782|0.8323|0.8281|0.8237|0.8451|0.8467|0.8583|0.8667|0.8467|0.8667|0.9000|0.9000|0.9200|0.9267|0.9167|
>
> **Table: Llama-3.1-8B-Instruct Reason**
>
> |Sample|1|5|10|15|20|25|30|35|40|45|50|55|60|65|70|80|90|95|
> |--------|------|------|------|------|------|------|------|------|------|------|------|------|------|------|------|------|------|-------|
> |Budget|14.0|56.67|110.0|163.33|216.67|270.0|323.33|376.67|430.0|483.33|536.67|590.0|643.33|696.67|750.0|856.67|963.33|1016.67|
> |Accuracy|0.7741|0.7860|0.7948|0.8022|0.8140|0.8247|0.8242|0.8188|0.8145|0.8147|0.8167|0.8161|0.8156|0.8240|0.8244|0.8149|0.8120|0.8159|
>
> **Table: Llama-3.1-70B-Instruct Reason**
>
> |Sample|1|5|10|15|20|25|30|35|40|45|50|55|60|65|70|80|90|95|
> |--------|------|------|------|-------|-------|------|-------|-------|------|-------|-------|------|-------|-------|------|-------|-------|-------|
> |Budget|138.0|511.33|978.0|1444.67|1911.33|2378.0|2844.67|3311.33|3778.0|4244.67|4711.33|5178.0|5644.67|6111.33|6578.0|7511.33|8444.67|8911.33|
> |Accuracy|0.7608|0.7960|0.8285|0.8312|0.8272|0.8261|0.8346|0.8416|0.8471|0.8494|0.8476|0.8422|0.8357|0.8312|0.8294|0.8304|0.8255|0.8232|
>
> **Table: Search Time and Performance across different methods on TextVQA with budget 1000. \* means failure to get the optimal configuration.**
>
> |Method|SearchTime(hrs)|Performance|
> |---------|-----------------|-----------|
> |Random|205*|0.90|
> |Bayes|50*|0.83|
> |MLCopilot|70*|0.90|
> |AgentHPO|245*|0.90|
> |**Ours**|**65**|**0.93**|
>
> **Key Takeaways:**
>
> * Test-time scaling boosts performance in multimodal tasks.
> * Under budget ≤ 500, **small models (e.g., 8B)** outperform **larger ones (e.g., 70B)**.
> * Our method can generalize to multimodal tasks.
>
> > W3 The FLOPs-based budget conversion assumes linear scaling relationships that may not hold across all model architectures and task types. Also, the choice of base configuration (3B model, 128 prompt tokens, 64 generation tokens) seems arbitrary and may bias results.
>
> We justify the linear budget scaling assumption in Appendix A.2. Intuitively, FLOPs scale linearly with model size $k$ and token length $m$—i.e., $k \times m$ increase leads to proportional compute growth.
>
> The base config (3B, 128 prompt + 64 gen tokens) was chosen to reflect realistic deployment. The 3B size typifies SLMs, and the token lengths were empirically verified across all used benchmarks and task types to meet minimal needs. This compact yet sufficient setup serves as a standard unit for flexible budget allocation.
>
> All methods use the same base config for fair and unbiased comparison. Though other base configs may yield different absolute budget values, our goal is to optimize allocation under a fixed total. We pick a minimal, representative unit—like choosing km vs. miles as a distance metric—valid as long as consistently applied.
>
> > W4 It seems only GPT-o3-mini is used as the agent LLM, the paper could benefit from applying the method on more LLMs to show its generalizability.
>
> We appreciate the reviewer’s suggestion to evaluate the generalizability of our method across different agent LLMs.
>
> In our main experiments, we used **GPT-o3-mini** as the search agent due to its strong reasoning and planning capabilities. However, we agree that it is important to assess whether our proposed method can adapt to other large language models. To address this, we conducted additional experiments using **DeepSeek-Chat** and **LLaMA 3.3-70B** as alternative agent LLMs. The results are as follows:
>
> **Table: Search Time on 2Wiki Dataset (in hours)**, -- means failure to find the best configuration
>
> |Method|GPT-o3-mini|DeepSeek-R1|Llama3.370B|
> |------|-----------|-----------|-----------|
> |LLM_ZS|12.5|12.5|--|
> |MLCopilot|12.5|12.5|--|
> |AgentTTS|8.3|16|--|
> |**Ours**|**2.5**|**3**|--|
>
> Experiments show that both our method and the baselines generalize well to DeepSeek-R1 but fail to transfer to Llama 3.3 70B. We attribute this to the search procedure’s reliance on the reasoning and planning capabilities of the LLM used for search.
>
> We hope our clarifications resolve the reviewer's concerns. Let us know if more clarification is needed.

---

> ### Author Response · Authors · 2025-08-05
>
> Dear Reviewer,
> As the rebuttal discussion deadline approaches, we would like to check if you have any further questions, we would be happy to provide additional clarification. If everything is clear, we kindly ask you to consider raising the rating based on our response.
> Thanks, The Authors

---

> > ### Comment · Reviewer_NhkM · 2025-08-07
> >
> > Thank you for your detailed responses. I've updated the score accordingly.

---

### Official Review · Reviewer_ZDCu · 2025-07-02

**Clarity:** 3
**Significance:** 2
**Originality:** 3
**Rating:** 4
**Confidence:** 3

**Summary:**

This paper introduces AgentTTS, a framework designed to solve the complex and highly practical problem of test-time compute-optimal scaling for multi-stage tasks. The core challenge it addresses is how to best allocate a limited computational budget across a sequence of subtasks—like in a retrieval-then-generation system—to maximize the overall performance. The authors first establish three key empirical insights from pilot experiments: that different subtasks have distinct model preferences, that performance returns diminish with added compute, and that budget choices in early stages impact later ones. AgentTTS uses an LLM-powered agent that leverages these insights to autonomously and iteratively search for the best combination of models and budget allocations. The agent interacts with a task environment, gets feedback, and refines its strategy to efficiently find a near-optimal configuration. Results show AgentTTS outperforms baselines in efficiency and robustness.

**Questions:**

As mentioned in the Weakness.

**Ethical Concerns:**

["NO or VERY MINOR ethics concerns only"]

**Limitations:**

As mentioned in the Weakness.

**Paper Formatting Concerns:**

Not found

**Quality:**

3

**Strengths And Weaknesses:**

Strengths：
1. The paper effectively targets a new problem of compute-optimal scaling in multi-stage complex tasks, which is highly relevant to real-world applications where tasks involve multiple stages with diverse model requirements.
2. The authors conduct experiments on models of various sizes （from 3B to 72B）and types (Qwen, Llama), providing a thorough evaluation of the proposed method across different model capabilities.
3. The authors have done a great job of validating their method across a wide range of tasks, from question answering to automated software development, using six different datasets.

Weaknesses：
1. Despite the novel problem proposed by the authors, the innovation of the method itself should be further elaborated.
2. Lacks in-depth theoretical analysis of the proposed method.
3. The paper focuses on optimizing the inference budget, but the search process itself has a computational cost. A more direct discussion about this trade-off would be beneficial. For instance, how much deployment is needed to justify the upfront cost of the search?

---

> ### Author Rebuttal · Authors · 2025-07-31
>
> We sincerely appreciate the reviewer’s careful evaluation and valuable comments on innovation, in-depth analysis of the proposed method, and the computational cost during searching. We provide detailed responses to each issue raised below.
>
>
>
> > W1 Despite the novel problem proposed by the authors, the innovation of the method itself should be further elaborated.
>
> We appreciate the reviewer’s interest in further understanding the innovation of our proposed method.
>
> The novelty of our method is closely tied to the three core insights presented in the paper, which directly address the unique challenges posed by **multi-stage complex tasks within a fixed budget**: (1)**Model and number of sampling across subtasks**: Each subtask may require a different model based on its reasoning complexity and domain. Our method explicitly considers how to identify the most suitable model for each subtask. (2) **Optimal sampling budget varies by subtask**: There exists an ideal number of samples for each subtask. Our framework is designed to discover this optimal per-subtask sampling rate. (3) **Global budget constraints create trade-offs**: Given a fixed compute budget, it is often impossible to allocate the optimal budget to all subtasks. Thus, we must prioritize subtasks and allocate resources in a way that **maximizes overall task performance**.
>
> To address these intertwined challenges, we propose a **LLM-agent-based framework that autonomously searches for compute-optimal allocations** that leverages both (i) the planning capability of LLMs and (ii) our proposed insights as **informed priors**. During the search process, the agent receives performance feedback and iteratively improves its configurations based on accumulated experience, leading to **more efficient search** and **superior configurations** compared to prior baselines. Thus, the **innovation lies not only in the LLM-agent-based framework** but in how it is ***informed and guided*** by empirically validated insights that are specific to **multi-stage, budget-constrained tasks**.
>
> We will make this connection between insights and algorithmic design clearer in the revised manuscript.
>
>
>
>
>
> > W2 Lacks in-depth theoretical analysis of the proposed method.
>
> Thank you to the reviewer for highlighting the lack of in-depth theoretical analysis of our search method. In response, we provide a complete theoretical analysis below, starting from the basic notations and culminating in a justification for why an insights-informed search leads to greater efficiency.
>
>
>
> **Notation.**
> Let $\mathcal{T} = [T_1, \dots, T_N]$ be the subtasks. For subtask $i$, let $\mathcal{M_i}$ be the model set and $M_i \in \mathcal{M_i}$  a choice.
> Let $\mathbf{B} = (B_1, \dots, B_N)$ be integer budgets with $\sum_i B_i \le B$ (total budget $B \in \mathbb{N}$).
> A configuration (trial) is $c = (M_1, B_1, \dots, M_N, B_N) \equiv (M, \mathbf{B})$. Let $F(M, \mathbf{B})$ denote the expected task score. AgentTTS seeks $c^\star = \arg\max_{(M, \mathbf{B})} F(M, \mathbf{B})$.
>
> **Assumptions (Insights 1–3 formalized).**
>
> - **A1 (Subtask-specific model preference).**  For each $i$ there exists a *dominant* model $M_{i}^* \in \mathcal{M_i}$ and a budget interval $\mathcal{I_i} \subseteq \{0, \dots, B\}$ such that for any fixed $ ( M_{-i}, \mathbf{B_{-i}}) $ and all $B_{i}  \in \mathcal{I_{i}}$,
> $$
> F(M_{i}^* , B_{i}; M_{-i}, \mathbf{B_{-i}}) - \max_{M_{i} \ne M_{i}^*} F(M_{i}, B_{i}; M_{-i}, \mathbf{B_{-i}}) \ge \Delta_{i} > 0.
> $$
>
> - **A2 (Unimodality in $B_i$).**  For fixed $M_i$ and $(M_{-i}, \mathbf{B}_{-i})$, the function $B_i \mapsto F(M, \mathbf{B})$ on $\{0, \dots, B\}$ is unimodal with a unique maximizer $B_i^\star(M)$.
>
> - **A3 (Bounded coupling).**  For fixed $M$, define the best-response map $T: \{0, \dots, B\}^N \to \{0, \dots, B\}^N$ by $
>   [T(\mathbf{B})]_i = \arg\max_b F(M, \mathbf{B} \mid B_i = b).
>   $ There exists $\rho \in (0, 1)$ such that $
>   \|T(\mathbf{B}) - T(\mathbf{B}')\|_2 \le \rho \, \|\mathbf{B} - \mathbf{B}'\|_2 \quad \text{for all } \mathbf{B}, \mathbf{B}'.
>   $
>
> **Initialization (Model elimination; Insight 1).**  Compare models within each $\mathcal{M}_i$ under aligned conditions (same $B_i \in \mathcal{I}_i$, others fixed).  By A1, the dominant model $M_i^*$ is separated by a margin $\Delta_i$, so selecting the top model per subtask is correct.  Keeping only a small shortlist $\widetilde{\mathcal{M}}_i$ that contains the winner reduces the search from $
> \prod_i |\mathcal{M}_i| \quad \text{to} \quad \prod_i |\widetilde{\mathcal{M}}_i|.
> $
>
> **Budget Localization (per subtask; Insight 2).**  With $M$ fixed (after elimination), $B_i \mapsto F(M, \mathbf{B})$ is unimodal (A2).  A simple bracket-and-zoom (e.g., ternary or golden-section on integers) halves the budget interval each round.  After $R = \mathcal{O}(\log(B/\varepsilon))$ rounds, the bracket length is $\le \varepsilon$, yielding $\widehat{B}_i$ with $
> |\widehat{B}_i - B_i^\star(M)| \le \varepsilon.
> $ Hence, the number of evaluations per subtask for budget refinement satisfies $
> m_i = \mathcal{O}(\log(B / \varepsilon)).
> $
>
> **Coordination and Convergence (Insights 3).** Fix $M \in \prod_i \widetilde{\mathcal{M}}_i$ and let $\mathbf{B}^\diamond$ be the unique fixed point of $T$ (A3).
> If AgentTTS updates budgets via (exact or close-to-exact) best responses, the error  $
> \mathbf{e}^{(t)} = \mathbf{B}^{(t)} - \mathbf{B}^\diamond
> $
> contracts geometrically:  $
> \|\mathbf{e}^{(t)}\|_2 \le \rho^t \, \|\mathbf{e}^{(0)}\|_2.
> $ The Archive retains the best-seen configuration, so the recorded best score is non-decreasing across iterations.
>
>
>
> **Why Insight 3 Accelerates Convergence:**
> A **naïve update** tweaks subtask $i$ while pretending the rest are fixed. Then move to subtask $j$ and tweak it, which **changes the conditions that made the last tweak to $i$ look good**. The result is that when come back to $i$, we often need to move it **back toward where it was**—so that we “undo” part of your previous change. This behaves like a weaker contraction with factor $\rho_{\text{naive}}$ (closer to 1).
>
> In constrast, Insight 3 lets the LLM *group and update strongly coupled subtasks together* (block best responses),  better approximating the full map $T$ and yielding a smaller factor $\rho_{\text{eff}} < \rho_{\text{naive}}$. Therefore, the insight 3 helps accelerate convergence.
>
>
>
>
> > W3 The paper focuses on optimizing the inference budget, but the search process itself has a computational cost. A more direct discussion about this trade-off would be beneficial. For instance, how much deployment is needed to justify the upfront cost of the search?
>
> Thank you for raising this important point. We agree that analyzing the trade-off between the computational cost of the search process and the resulting inference-time performance is essential for assessing our method.
>
> To address this, we now include a practical estimate illustrating how quickly the search cost can be amortized through downstream inference improvements under typical deployment scenarios. For instance, on the 2Wiki dataset, our method's search process requires approximately 2.5 GPU-hours. Given that our optimized configuration achieves a 10–20% increase in test accuracy compared to baseline methods at a fixed inference budget, this initial computational investment is offset after only a few hundred test-time executions—a frequency commonly observed in real-world batch deployments. This scenario is analogous to fine-tuning models for specific downstream tasks, where investing several GPU-hours upfront significantly enhances model performance during inference.
>
> We believe this analysis helps clarify when the upfront search effort is worthwhile.
>
>
>
> Thank you again for your thoughtful review. We welcome any additional feedback.

---

> > ### Comment · Reviewer_ZDCu · 2025-08-09
> >
> > Thank you for your detailed rebuttal. I appreciate the time and effort you have put into addressing the points raised in my review. Your explanations have successfully resolved most of my concerns. I will keep my socre.

---

### Official Review · Reviewer_bAdt · 2025-07-03

**Clarity:** 2
**Significance:** 2
**Originality:** 3
**Rating:** 4
**Confidence:** 3

**Summary:**

This paper addresses the challenge of test-time compute-optimal scaling in multi-stage complex tasks, where existing methods for single-stage tasks fall short due to the combinatorial search space and interdependent subtasks. The authors first conduct pilot experiments, deriving three key insights. Guided by these, they propose AgentTTS, an LLM-agent framework that iteratively searches for optimal compute allocations via feedback-driven interactions. AgentTTS leverages an LLM to generate candidate trials, update guidelines based on performance feedback, and adapt to subtask dependencies. Experiments show that AgentTTS outperforms traditional and LLM-based baselines in search efficiency

**Questions:**

See above

**Ethical Concerns:**

["NO or VERY MINOR ethics concerns only"]

**Final Justification:**

My current assessment leans toward a score of 3.5. While most of my concerns have been addressed, I find the definition of the budget unit remains unclear. This aspect appears to unnecessarily complicate the paper's readability while contributing minimally to either the methodology or results. I recommend that the AC also consider the clarity of writing when making their final decision.

**Limitations:**

yes

**Quality:**

2

**Strengths And Weaknesses:**

**Strengths:
1. The pilot study derives three key insights from comprehensive experiments, showing that different subtasks prefer distinct models, compute scaling has diminishing returns, and earlier budgets affect downstream tasks. These insights ground the framework in real-world LLM behavior.
2. AgentTTS reduces search time by up to 80% compared to baselines while matching or exceeding performance, demonstrating practical utility for real-world deployment.
3. By focusing on multi-stage task compute optimization, the paper addresses a critical gap in existing TTS research, which primarily targets single-stage tasks. This aligns with complex real-world scenarios like retrieval-then-generation QA and software development .

**Weaknesses:**
1. While Theorem 1 and the budget unit definition aim to standardize compute metrics, the experimental section rarely references these concepts, making their practical impact unclear. The complex formula may introduce unnecessary complexity without tangible benefits in results.
2. The three core insights, subtask model preferences, optimal budget saturation, and downstream dependency, are largely intuitive and supported by common LLM observations. While validated experimentally in this paper, they may lack the novelty expected for foundational "insights".
3. Inconsistent result presentation:
   - Figure 3 shows baselines approaching 0.8 EM on 2WikiMultiHopQA, but Table 1 lists AgentTTS's peak at 0.72. Why? I thought it was because Figure 3 was reporting performance on training set, but I read that Line 325 says that Figure 3 is also the performance on test set.
   - From Figure 3, it seems most of the methods have the same performance upper bound, why? What's the dashed line in each plot? It is also unclear for the caption potentially refers to the dashed line: "The best score is obtained from the optimal trial in a prior grid search and serves as the benchmark for all methods".
   - AgentTTS appears to still be improving at 50 trials on 2WikiMultiHopQA, raising questions about its performance with more iterations. Why not present the results with more trials?

---

> ### Author Rebuttal · Authors · 2025-07-31
>
> We thank the reviewer for the thoughtful and detailed feedback on the budget unit, the proposed insights, and others. We address the comments point-by-point below.
>
> > While Theorem 1 and the budget unit definition aim to standardize compute metrics, the experimental section rarely references these concepts, making their practical impact unclear. The complex formula may introduce unnecessary complexity without tangible benefits in results.
>
> Thank you for the thoughtful comment. We appreciate the opportunity to clarify the motivation and practical role of Theorem 1 and the budget unit definition.
>
> **Purpose of Theorem 1 and Budget Unit Definition:**
>
> The budget unit formulation in Theorem 1 is not intended to add unnecessary complexity, but rather to **standardize the notion of compute cost** across diverse tasks and models. In practice, different deployment scenarios may use varied cost metrics such as FLOPs, latency, token usage, or even monetary cost. Without a unified abstraction, comparisons between models would be inconsistent and task-specific.
>
> Our budget unit definition offers a **task- and model-invariant compute framework** that enables fair cross-task/model comparisons, supports any cost metric via normalization, and provides a consistent interface for budget-aware reasoning.
>
> **Practical Relevance in Experiments:**
>
> While the experimental section may not explicitly re-state Theorem 1, the concept is tightly integrated into both the **design of our method** and the **analysis of results**:
>
> 1. **Insight-driven analysis (Fig. 1)** relies on plotting performance as a function of budget, enabling meaningful comparison of models under equal compute budget.
>
> 2. **Method design (Eq. 5)** depends on a consistent budget unit to initialize candidate trials and perform model–sampling trade-offs during search.
>
> 3. **The LLM-based search agent** uses the budget computation formula internally to assess feasibility and guide its planning strategy during the search process.
>
> Thus, the theoretical framework directly informs the architecture of our search method and the interpretation of test-time scaling behavior.
>
>
>
>
>
> > W2 The three core insights, subtask model preferences, optimal budget saturation, and downstream dependency, are largely intuitive and supported by common LLM observations. While validated experimentally in this paper, they may lack the novelty expected for foundational "insights".
>
>
>
> We thank the reviewer for this comment. We agree that the three insights—**subtask-specific model preference**, **optimal budget saturation**, and **downstream dependency**—are partially aligned with emerging observations in LLM literature. However, we respectfully argue that their **generalization and algorithmic application** in our work offer substantial contributions beyond prior findings.
>
> **Relation to Prior Work:**
>
> Some individual components of our insights do echo trends observed in prior studies:
>
> - **Insight 1 (Subtask-specific model preference)** aligns with findings in [1], which shows that **easier samples benefit more from small models** under test-time scaling, while **harder ones require larger models**. However, this was shown primarily in single-task, single-stage, and limited task-type settings (mainly MATH dataset), whereas we generalize this to **multi-subtask pipelines** across six diverse datasets.
> - **Insight 2 (Unimodal budget saturation)** aligns conceptually with the saturation effect in [2], where performance plateaus under majority voting. However, [2] relies on a **verifier model** for weighted aggregation, whereas our **sampling-then-fusion** approach is verifier-free, making it more general and practical.
> - **Insight 3 (Downstream dependency)**, to the best of our knowledge, is **novel**. It emerges only in **multi-stage pipelines**, where upstream outputs influence downstream task difficulty and model choice. We are the first to systematically identify and formalize this inter-stage interaction, which significantly affects budget allocation dynamics.
>
> **Contribution and Practical Role:**
>
> While the insights may appear intuitive in isolation, our contribution lies in:
>
> 1. **Demonstrating their generality** by validating all three across **six datasets spanning four task types**: retrieval, question answering, summarization, and knowledge graph reasoning—far beyond the narrow scope of prior works.
>
> 2. **Using them to inform the design of our search algorithm**, which directly incorporates these principles to improve search efficiency. It yields **substantial empirical speed-ups** in identifying optimal configurations (see Table 1).
>
> [1] *Scaling LLM Test-Time Compute Optimally can be More Effective than Scaling Model Parameters*.
> [2] *Inference Scaling Laws: An Empirical Analysis of Compute-Optimal Inference for Problem-Solving with Language Models*.
>
> > Figure 3 shows baselines approaching 0.8 EM on 2WikiMultiHopQA, but Table 1 lists AgentTTS's peak at 0.72. Why? I thought it was because Figure 3 was reporting performance on training set, but I read that Line 325 says that Figure 3 is also the performance on test set.
>
> We appreciate the reviewer catching this inconsistency. There is a **typographical error in Line 325**, which mistakenly states that Figure 3 reports *test set* performance.
>
> **Correction**: *Fig. 3 shows the search trajectories of our method and the baselines for 325 test-time compute-optimal budget allocation on the training set.*
>
> > From Figure 3, it seems most of the methods have the same performance upper bound, why? What's the dashed line in each plot? It is also unclear for the caption potentially refers to the dashed line: "The best score is obtained from the optimal trial in a prior grid search and serves as the benchmark for all methods".
>
> As to why many methods appear to approach a same upper bound: this is **by design**. The core goal of our method (and baselines) is not to exceed this bound, but to **reach it efficiently** with fewer trials. Theoretically, any method (even random search) could reach the optimum if allowed infinite time—hence they share the same asymptotic limit.
>
> The dashed line in Figure 3 indicates the **best achievable performance** on the training set, which we use as a reference point to assess the search efficiency of all methods. Specifically:
>
> - This best score is obtained by conducting an approximated exhaustive grid search across the entire configuration space, which includes all possible combinations of model selections and sampling budgets for each subtask. Specifically, we enumerate all candidate configurations, execute them in the environment, and record their final task performance. The configuration that achieves the highest score is selected as the best. We will clarify this process more explicitly in the figure caption.
> - The purpose of this line is to serve as a **compute-optimal upper bound**—i.e., the highest training-set score obtainable with a fixed budget.
>
>
> > AgentTTS appears to still be improving at 50 trials on 2WikiMultiHopQA, raising questions about its performance with more iterations. Why not present the results with more trials?
>
> While Figure 3 shows a continued upward trend for AgentTTS on 2WikiMultiHopQA, we clarify that this behavior simply reflects **fine-grained fluctuations** near the optimal configuration. In practice, AgentTTS has already **found the optimal (or near-optimal) configuration** by that point. Extending the number of trials beyond 50 does not lead to meaningful gains, as the method has already converged to the compute-optimal region.
>
> We chose 50 trials as most methods can reach the optimum within this limit. AgentHPO typically uses 10 trials, so 50 is already a generous budget that provides ample opportunity to find the best configuration. Beyond this, the search cost becomes excessive—for example, exceeding 25 hours on the 2Wiki dataset. We will make this clearer in the revised version.
>
>
> We hope our clarifications resolve your concerns and demonstrate the robustness of our contributions. Please let us know if any further explanations are needed.

---

> > ### Comment · Reviewer_bAdt · 2025-08-06
> >
> > Thank you for your response. I've updated the score.

---

> ### Author Response · Authors · 2025-08-05
>
> Dear Reviewer,
> As the rebuttal discussion deadline approaches, we would like to check if you have any further questions, we would be happy to provide additional clarification. If everything is clear, we kindly ask you to consider raising the rating based on our response.
> Thanks, The Authors

---

### Official Review · Reviewer_8uj2 · 2025-07-07

**Clarity:** 2
**Significance:** 3
**Originality:** 3
**Rating:** 4
**Confidence:** 4

**Summary:**

The paper proposes AgentTTS, an LLM-guided agent that optimizes compute allocation across multi-stage NLP pipelines under test-time budgets. It leverages empirical insights such as diminishing returns and subtask dependencies to guide efficient search, outperforming multiple baselines on six benchmarks.

**Questions:**

Q1. Given that several reported gains are relatively small (e.g., 1–2 EM points), can you provide statistical significance estimates or confidence intervals for key results (perhaps through multiple seeds or bootstrap analysis)?

Q2. How were the HPO baseline methods adapted to handle the multi-stage compute-aware pipeline setting? Is there evidence that they are competitive under best-case tuning?

Q3. Have you considered or tested AgentTTS on pipelines involving other domains, modalities, or more complex, longer-horizon tasks? What might limit generalization to these settings?

Q4. Have you considered repeating your analysis for different temperature values? For example, this [paper](https://arxiv.org/abs/2504.12951) [1] shows that increasing the temperature value influences the sample efficiency of various reasoning methods.

[1] Potamitis et al. Are Retrials All You Need? Enhancing Large Language Model Reasoning Without Verbalized Feedback. https://arxiv.org/abs/2504.12951

**Ethical Concerns:**

["NO or VERY MINOR ethics concerns only"]

**Final Justification:**

I appreciate that the authors have provided detailed responses to my questions; however, as things stand, I would like to maintain my original assessment of the paper. Should the paper get accepted, I would also appreciate it if the AC urges the authors to add the results from their response in the final version of their paper.

Finally, as stated in my response to the authors, I would appreciate it if the authors showcase their findings on tasks beyond "question answering" and from diverse domains.

**Limitations:**

Yes.

**Quality:**

4

**Strengths And Weaknesses:**

### Strengths

S1. Open source code.

S2. Comprehensive evaluation on six datasets.

S3. Clear empirical insights are discussed in the introduction that guide the development of the method.

S4. Strong empirical performance. Figure 3 shows the sample efficiency of AgentTTs while Table 1 (partly) shows its cost efficiency in terms of search time.

### Weaknesses

W1. Table 1: Since there are a lot of unavailable results, it’s hard to understand the table completely. For example, in the Taskbench and Chatdev case, there’s no search time included for any of the baseline search times, even though some of them achieve the same score as AgentTTs.

W2. No error bars are reported, which raises a question regarding the robustness of the proposed method.

W3. Domain coverage. Even though the authors benchmark their method against multiple baselines and different benchmarks, the domain that the benchmarks include is fairly limited. The used benchmarks focus on QA-style pipelines; however, I would like to see some results on different domains as well, such as math, and more complex benchmarks such as planning.

---

> ### Author Rebuttal · Authors · 2025-07-31
>
> We appreciate the reviewer’s valuable comments. Below, we respond to each point in detail.
>
> > W1 Table 1: Since there are a lot of unavailable results, it’s hard to understand the table completely. For example, in the Taskbench and Chatdev case, there’s no search time included for any of the baseline search times, even though some of them achieve the same score as AgentTTs.
>
> In the orginal table, we omitted search times for methods that failed to find optimal configs on the trainng set to focus on those with strong **generalization** and **efficiency**. Reported test results reflect the generalization of training-found optimal or near-optimal configs.
>
> To address the concern, we have included the previously missing search times in a revised version of Table below. Entries marked with an asterisk (*) indicate cases where the method did not find the best-performing configuration in the training set; in such cases, the search time corresponds to the best configuration the method could find, which often leads to suboptimal test-set performance.
>
> **Table:** Comparison of search time (in hours) and test-set performance across datasets. * indicates the method did not find the best trial in the training set.
>
> |Method|2WikiTime|2WikiEM|2WikiEM(90%CI)|HotpotTime|HotpotEM|CWQTime|CWQEM|WebQSPTime|WebQSPEM|TaskbenchTime|Taskbenchp-F1|ChatDevTime|ChatDevCons.|
> |------|---------|-------|---------------|-----------|--------|--------|------|------------|----------|---------------|---------------|-------------|---------------|
> |Random|24.3*|0.66|0.656(0.624,0.688)|16.4*|0.71|37.4*|0.76|10.0*|0.86|14.6*|0.40|18.4*|0.74|
> |BO|3.5*|0.60|0.604(0.574,0.635)|40.5*|0.71|23.8*|0.76|10.0*|0.85|14.6*|0.52|4.6*|0.75|
> |LLM_ZS|12.5|0.70|0.677(0.625,0.729)|13.1*|0.71|55.3|0.76|37.7|0.89|114.0*|0.49|2.0*|0.74|
> |MLCopilot|12.5|0.70|0.698(0.680,0.716)|46.3|0.72|48.4|0.78|20.8|0.88|23.4*|0.53|20.9*|0.75|
> |AgentHPO|8.3|0.70|0.698(0.680,0.716)|36.3|0.74|37.4|0.78|48.1|0.89|43.8*|0.49|16.9*|0.74|
> |**Ours**|**2.5**|**0.72**|**0.724(0.695,0.752)**|**17.5**|**0.74**|**11.1**|**0.78**|**7.8**|**0.89**|**64.3**|**0.53**|**14.3**|**0.75**|
>
>
> > W2 No error bars are reported, which raises a question regarding the robustness of the proposed method.
> >
> > Q1. Given that several reported gains are relatively small (e.g., 1–2 EM points), can you provide statistical significance estimates or confidence intervals for key results (perhaps through multiple seeds or bootstrap analysis)?
>
> Since the weakness and question overlap, we respond jointly. We ran 5 repeated searches on 2Wiki with different seeds and report **mean ± 90% CI** in the table above to assess robustness.
>
> The results confirm that while there is natural variance across runs, our method consistently discovers high-quality configurations. On average, the configurations identified by our approach achieve **superior generalization performance** on the test set compared to baselines, further supporting the robustness and effectiveness of our search strategy.
>
> > W3. Domain coverage. Even though the authors benchmark their method against multiple baselines and different benchmarks, the domain that the benchmarks include is fairly limited. The used benchmarks focus on QA-style pipelines; however, I would like to see some results on different domains as well, such as math, and more complex benchmarks such as planning.
> >
> >  Q3. Have you considered or tested AgentTTS on pipelines involving other domains, modalities, or more complex, longer-horizon tasks? What might limit generalization to these settings?
>
> Both the weakness and question concern the same point, so we address them together. While our main experiments use QA-style pipelines, **our method is domain-agnostic** and applies to any multi-stage tasks. It assumes modular we have subtasks and model choices, which generalize across domains—for example, geography VQA can be split into image parsing (e.g., maps) and reasoning (e.g., climate), each handled by specialized models.
>
> 1. **Why math and planning tasks were not included initially**
>
> We also tried mathematics or planning datasets but not included in our main evaluation because **existing benchmarks in those domains are inherently single-stage**. For example:
>
> - **Math tasks** such as *MATH*, and *MathQA* typically require long-form generation but do not involve discrete multi-stage components.
> - **Planning tasks** (e.g., ALFWorld) are often benchmarked via environment interactions rather than modular subtask execution.
>
> Our framework is designed for structured reasoning with budget allocation across dependent subtasks. Thus, without existing multi-stage decompositions, math and planning tasks fall outside the scope of the current setting.
>
> 2. **New domain experiment: Multimodal reasoning on TextVQA**
>
> To address the concern about generalization to other domains/modalities, we adapted the **TextVQA** dataset into a two-stage pipeline: **Stage 1**: Image understanding; **Stage 2**: Question-specific reasoning based on image understanding.
>
> We then evaluated the **search efficiency** for multiple models in this multimodal setting. Results are shown below.
>
> **Table: Search Time and Performance across different methods on TextVQA with budget 1000. \* means failure to get the optimal configuration.**
>
> |Method|SearchTime(hrs)|Performance|
> |------|----------------|-----------|
> |Random|205*|0.90|
> |Bayes|50*|0.83|
> |MLCopilot|70*|0.90|
> |AgentHPO|245*|0.90|
> |**Ours**|**65**|**0.93**|
>
> The experimental results show our method could generalize to the multi-modal task.
>
> 3. **Test-time scaling on MATH dataset**
>
> To address the request, We **run MATH experiments** with Qwen2.5 (3B, 72B) to test how **test-time scaling** affects open-ended math reasoning. Accuracy on 500 samples is shown below with **Best-of-N aggregation**.
>
> **Table: Qwen 72B test-time scaling on MATH test set with 500 samples using Best-of-N aggregation**
>
> |RepeatCount|Budget|Accuracy|
> |-----------|------|--------|
> |1|214|0.676|
> |2|406|0.708|
> |3|598|0.724|
> |4|790|0.730|
> |5|982|0.738|
> |6|1174|0.740|
> |7|1366|0.742|
> |8|1558|0.742|
> |9|1750|0.744|
> |10|1942|0.744|
>
>
> **Table: Qwen 3B test-time scaling on MATH test set with 500 samples using Best-of-N aggregation**
>
> |RepeatCount|Budget|Accuracy|
> |-----------|------|--------|
> |1|7|0.506|
> |10|79|0.640|
> |20|159|0.670|
> |30|239|0.670|
> |40|319|0.676|
> |50|399|0.680|
> |60|479|0.684|
> |70|559|0.682|
> |80|639|0.684|
> |90|719|0.686|
>
> Qwen2.5-72B costs 214 units per gen; 3B costs 7. Though 3B can beat 72B with >50 repeats, it does so at **higher total cost**. Thus, under fixed budgets, **72B is more cost-effective** for MATH.
>
> > Q2. How were the HPO baseline methods adapted to handle the multi-stage compute-aware pipeline setting? Is there evidence that they are competitive under best-case tuning?
>
> **How HPO Methods Were Adapted:**
>
> LLM-based HPO methods, such as AgentHPO, were originally developed to tune hyperparameters for machine learning tasks by having an LLM iteratively propose candidate configurations based on task descriptions and observed feedback. In our setting, **test-time scaling configuration search closely parallels HPO**, with the distinction that the parameters to be optimized are no longer traditional hyperparameters, but rather **test-time scaling configurations**—i.e., model choices and generation budgets across multiple subtasks. This conceptual alignment motivated us to adopt HPO methods as baselines for our work.
>
> We adapted **AgentHPO** as follows:
> * Provide task description and config space to the LLM.
> * LLM generates **initial trials**, evaluated for feedback.
> * Log trials with **rationale-style experiences** per AgentHPO.
> * LLM uses past feedback to **plan next trials**.
> * Select **best training trial** for test evaluation.
>
> **Evidence of Competitiveness:**
>
> The effectiveness of LLM-based tuning is supported by the results reported in **Table 1** and **Figure 3** in the paper. In particular:
> - **AgentHPO is among the strongest HPO baselines**, frequently identifying near-optimal configurations in training.
> - Its performance serves as a robust foundation for evaluating our method.
>
> Our method builds on this strong baseline by incorporating three key insights (as described in Section 4), which improve **efficiency** of the search. Nonetheless, the baseline AgentHPO method itself is already capable and competitive under best-case tuning, reinforcing the relevance and strength of the comparison.
>
> > Q4. Have you considered repeating your analysis for different temperature values? For example, [1] shows that increasing the temperature value influences the sample efficiency of various reasoning methods.
>
> In our main experiments, we used a relatively **high temperature value of 0.7 or 0.9 by default**. However, we had not systematically varied the temperature values in earlier analysis.
>
> To address this concern, we conducted additional experiments on 2Wiki to evaluate the effect of temperature on test-time scaling. Specifically, we fixed the retriever to **Qwen2.5-72B** (generating a single retrieval result) and varied the number of samples and temperature values for the **QA model LLaMA 3 3B** on the **2Wiki dataset**. The results are:
>
> **Table: EM Scores at Different Temperatures on 2Wiki**
>
> |Samples|Temp0.1|Temp0.5|Temp0.9|
> |-------|--------|--------|--------|
> |1|0.70|0.64|0.66|
> |10|0.70|0.78|0.78|
> |20|0.72|0.76|0.78|
> |30|0.74|0.78|0.72|
> |40|0.74|0.76|0.78|
> |50|0.74|0.72|0.78|
> |60|0.74|0.78|0.80|
> |70|0.76|0.72|0.74|
> |80|0.76|0.76|0.78|
> |90|0.74|0.72|0.76|
>
> Key Observations:
> * **High temps (e.g., 0.9)** boost test-time scaling by increasing output diversity, aiding fusion with more samples; **low temps (e.g., 0.1)** suit single-sample cases.
> * This aligns with [1], showing **diverse outputs are key** for sampling-based reasoning.
>
> We hope our replies clarify your concerns. Let us know if you have further questions.

---

> ### Comment · Reviewer_8uj2 · 2025-08-06
> **Thanks for your thoughtful response**
>
> Dear Authors,
>
> Thanks a lot for your detailed and thoughtful response. Your response answers all of my questions but one. I would appreciate it if you could include all the new results and analyses along with the necessary discussions in the revised version of your manuscript.
>
> Regarding your response on the tasks, there are many mathematical reasoning and planning tasks that are naturally multi-step, such as "Game of 24", "Crossword", "Web Navigation", etc. Thus, I don't fully agree with your response in this regard. Further, I wonder why such tasks were not considered in your study. Thanks!
>
> Best,
> Reviewer 8uj2

---

> > ### Author Response · Authors · 2025-08-07
> >
> > Dear Reviewer 8uj2,
> >
> > Thank you for the thoughtful feedback. I acknowledge your point that many existing mathematical reasoning and planning tasks are inherently multi-step. However, our study focuses on a different class of problems: multi-stage tasks where each subtask requires distinct abilities or domain knowledge, making it necessary to dynamically select different models and test-time configurations across subtasks.
> >
> > In contrast, for tasks such as Game of 24 or Web Navigation, a single model with strong planning or reasoning capability can often be applied uniformly across all steps. In such cases, the test-time configuration tends to remain fixed throughout the task, and there is limited need for selective reconfiguration based on subtask-specific requirements. As a result, these tasks are less suited for evaluating test-time scaling strategies that adapt to heterogeneous subtask demands, which is the central focus of our work.
> >
> > Hope these clarifications address your concern! Please do not hesitate to reach out if you have any further questions or concerns.
> >
> > Best, The Authors

---

### Comment · Area_Chair_NGwC · 2025-08-03

Dear Reviewers,

Please take a moment to revisit the paper, evaluate the authors’ reponses, and confirm whether the updates address any concerns raised during your initial review. If you have already completed this step, thank you for your swift action. Otherwise, we appreciate you doing so as soon as possible. Thank you again for your valuable time and contributions!

Thanks，

AC

---

### Note · Authors · 2025-08-12

We sincerely thank all reviewers for their time and constructive feedback. We have addressed each concern in detail, including additional experiments on the new-modal task and the MATH domain dataset, as well as a thorough theoretical analysis of our insights and proposed methods. We hope our responses help clarify our insights and the AgentTTS framework, and that they are valued in the final evaluation.

---

### Decision · Program_Chairs · 2025-09-17

**Decision:**

Accept (poster)

**Comment:**

The paper frames test-time scaling (TTS) for multi-stage pipelines as a compute-optimal allocation problem over (i) model choice per subtask and (ii) per-subtask sampling budgets under a fixed overall budget. Pilot studies yield three recurring patterns: (I) subtasks prefer different models, (II) returns w.r.t. repeats are unimodal / saturating, and (III) earlier-stage budget affects downstream difficulty and optimal allocations. Building on these, AgentTTS is an LLM-agent that iteratively proposes configurations, reads feedback, updates guidelines, and converges to near-optimal allocations with markedly fewer trials than baselines. Across six datasets (2Wiki, HotpotQA, CWQ, WebQSP, TaskBench, ChatDev) and models from 3B to 72B, AgentTTS matches or exceeds prior methods while reducing search time (often substantially).

Strengths:

- Compute-optimal TTS in multi-stage workflows is both under-explored and practically important.

- The three empirically backed patterns are baked into the agent’s search policy and improve sample efficiency.

- Six datasets, multiple model scales and consistent search-time reductions with comparable or better accuracy.

Weaknesses:

- The “budget unit” and FLOPs normalization remain under-explained in the main text; a train/test mismatch around Fig. 3 required rebuttal clarification.

- The three insights, while useful, are partly intuitive and the theoretical support is sketch-level and appears mainly in rebuttal.

- Core experiments skew QA-style and multimodal and math results appear only post-rebuttal and are limited.

- Success relies on a capable planner LLM and cross-agent robustness was only lightly probed (e.g., DeepSeek ok, Llama-3.3-70B struggled).

- Error bars/seed variability are not pervasive across all tables and some baselines’ search-time entries were initially missing.

Decision:

This paper is the first solid treatment of compute-optimal, multi-stage TTS with interdependent subtask choices. It reliably reaches near-optimal configs faster than strong LLM-HPO baselines. Though not exhaustive, new TextVQA and MATH results, plus cross-agent notes, show the approach is not QA-only. Authors resolved presentation issues, filled missing baselines/stats, and provided theory intuition and cost amortization.

Discussion:

- 8uj2 (score 4): Asked for error bars, baseline adaptation, tasks beyond QA, temperature effects. Authors added CI on 2Wiki (5 seeds), clarified HPO adaptations, provided TextVQA and MATH TTS studies, and a temperature grid; reviewer kept a 4.

- bAdt (score 4): Concerned budget unit/Theorem 1 relevance and result inconsistencies (Fig. 3 vs Table 1; dashed-line meaning). Authors clarified Fig. 3 is training-set trajectories, explained the upper-bound dashed line and how the budget abstraction is used throughout, plus why 50 trials suffice.

- ZDCu (score 4): Wanted clearer innovation, theory depth, and search-cost trade-off. Authors tied insights to algorithmic blocks, gave a convergence/efficiency sketch, and provided an amortization argument (e.g., ~2.5 GPU-hrs search).

- NhkM (score 4): Asked for principled theory, applicability limits, FLOPs linearity & base config choice, and agent-LLM diversity. Authors offered a formalization of the insights, defended FLOPs scaling, and added agent variants (DeepSeek ok; Llama-3.3-70B struggled).